# DECOUPLED MIXUP FOR DATA-EFFICIENT LEARNING

## ABSTRACT

Mixup is an efficient data augmentation approach that improves the generalization of neural networks by smoothing the decision boundary with mixed data. Recently, *dynamic* mixup methods have improved previous *static* policies effectively (*e.g.*, linear interpolation) by maximizing salient regions or maintaining the target in mixed samples. The discrepancy is that the generated mixed samples from *dynamic* policies are more instance discriminative than the *static* ones, *e.g.*, the foreground objects are decoupled from the background. However, optimizing mixup policies with dynamic methods in input space is an expensive computation compared to static ones. Hence, we are trying to transfer the decoupling mechanism of dynamic methods from the data level to the objective function level and propose the general decoupled mixup (DM) loss. The primary effect is that DM can adaptively focus on discriminative features without losing the original smoothness of the mixup while avoiding heavy computational overhead. As a result, DM enables *static* mixup methods to achieve comparable or even exceed the performance of *dynamic* methods. This also leads to an interesting objective design problem for mixup training that we need to focus on both smoothing the decision boundaries and identifying discriminative features. Extensive experiments on supervised and semi-supervised learning benchmarks across seven classification datasets validate the effectiveness of DM by equipping it with various mixup methods.

## 1 INTRODUCTION

Deep Learning has become the bedrock of modern AI for many tasks in machine learning (Bishop, 2006) such as computer vision (He et al., 2016; 2017), natural language processing (Devlin et al., 2018). Using a large number of learnable parameters, deep neural networks (DNNs) can recognize subtle dependencies in large training datasets to be later leveraged to perform accurate predictions on unseen data. However, models might overfit the training set without constraints or enough data (Srivastava et al., 2014). To this end, regularization techniques have been deployed to improve generalization (Wan et al., 2013), which can be categorized into data-independent or data-dependent ones (Guo et al., 2019). Some data-independent strategies, for example, constrain the model by punishing the parameters' norms, such as weight decay (Loshchilov & Hutter, 2017). Among data-dependent strategies, data augmentations (Shorten & Khoshgoftaar, 2019) are widely used.

Mixup (Zhang et al., 2017; Yun et al., 2019), a data-dependent augmentation technique, is proposed to generate virtual samples by a linear combination of data pairs and the corresponding labels with the mixing ratio $\lambda \in [0, 1]$. DNNs trained with this technique are typically more generalizable and calibrated (Thulasidasan et al., 2019), whose prediction accuracy tends to be consistent with confidence. The main reason is that mixup heuristically smooths the decision boundary to improve

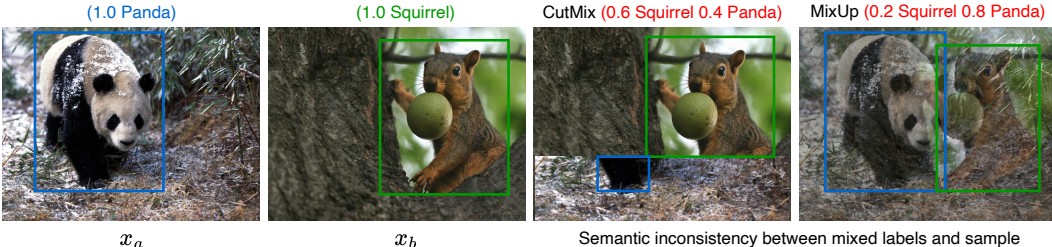

Figure 1: Illustration of the problem of label mismatch. The red mixed labels are the ground truth.

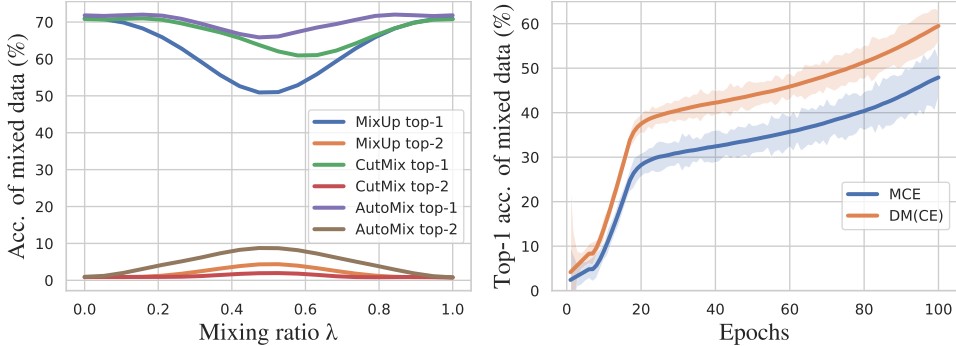

Figure 2: Experimental overviews of the label mismatch issue. Compared with *static* policies like Mixup and CutMix, the *dynamic* method AutoMix significantly reduces the difficulty of mixup classification and alleviates the label mismatch problem by providing more reliable mixed samples, but also brings a large computational overhead. **Left**: Top-1 and Top-2 accuracy of mixed data on ImageNet-1k with 100 epochs. Prediction is counted as correct if the Top-1 prediction belongs to $\{y_a, y_b\}$; prediction is counted as correct if the Top-2 predictions are equal to $\{y_a, y_b\}$. **Right**: Taking Mixup as an example, decouple mixup cross-entropy (DMCE) significantly improves training efficiency by alleviating the label mismatch issue from the perspective of designing a loss function.

the overall robustness by regressing the mixing ratio $\lambda$ in mixed labels. However, a completely handcrafted mixing policies (Verma et al., 2019; Uddin et al., 2020; Harris et al., 2020) (referred to as *static* methods) may result in a mismatch issue between the mixed labels and the mixed samples, which leads to problems such as instability or slow convergence, namely label mismatch (Liu et al., 2022), as shown in Figure 1. To mitigate this problem, a line of time-consuming mixup methods are proposed to improve mixing policies to generate object-aware virtual samples by optimizing discriminative regions in the data space (Uddin et al., 2020; Kim et al., 2020; 2021) (referred to as *dynamic* methods). For example, in Figure 2 *left*, the *dynamic* methods like AutoMix (Liu et al., 2022) improve mixup accuracy significantly. The basic idea is to decouple the foreground from the background and mix their corresponding features to avoid label mismatch. However, these data-level decoupling methods require a complex deployment and an additional 1.5 times the training time as static methods, which may violate the mixup augmentations' ease of use and lightness. Thus, leaving aside the design of a new mixup policy, a new question raises that **can we design a loss function for mixup that takes into account both the smoothness of mixup and the discriminatory of instances without heavy computation?** From a different perspective, we first regard the label-mismatched samples as hard mixed samples. In other words, even though there are not sufficient features in the mixed sample, we still expect the model to have the ability to mine these hard features to make confident predictions. Therefore, we make full use of mixed samples without introducing additional computational effort to achieve data-efficient mixup training.

Motivated by this intuition, we introduce Decoupled Mixup (DM), a mixup objective function for explicitly leveraging the target-relevant information of mixed samples without losing original smoothness. Based on the standard cross-entropy loss, an extra decoupled term is introduced to enhance the ability to mine underlying discriminative statistics in the mixed sample by independently computing the predicted probabilities of each mixed class. As a result, DM can further emphasize the contribution of each involved class in mixup to explore the efficient usage of existing data. Extensive experiments demonstrate that DM achieves data-efficiency training on supervised and semi-supervised learning benchmarks. Our contributions are summarized below:

- Unlike those static mixup policies that suffer from the label mismatch problem, we propose DM, a mixup objective of mining discriminative features while maintaining smoothness.

- Our work contributes more broadly to understanding mixup training: it is essential to focus not only on the smoothness by regression of the weight of mixing but also on discrimination by encouraging the network to give a highly confident prediction when the evidence is clear.

- The proposed DM can be easily generalized to semi-supervised learning by a minor modification. By leveraging the unlabeled data efficiently, it can reduce the conformation bias and significantly improve overall performance.

- Comprehensive experiments on various tasks verify the effectiveness of DM, *e.g.*, DM-based *static* mixup policies achieve a comparable or even better performance than *dynamic* methods without the extra computational overhead.

## 2 PRELIMINARIES

### 2.1 $\lambda$ WEIGHTED CROSS-ENTROPY UNDERUTILIZES MIXUP

Firstly, we dive into analyzing the generation of mixed samples and the original loss function. Let us define $y \in \mathbb{R}^C$ as the ground-truth label with $C$ categories. For data point $x \in \mathbb{R}^{\mathcal{W} \times \mathcal{H} \times \mathcal{C}}$ whose embedded representation $z$ is obtained from the model $M$ and the predicted probability $p$ can be calculated through a Softmax function $p = \sigma(z)$. Given the mixing ratio $\lambda \in [0, 1]$ and $\lambda$ related mixup mask $H \in \mathbb{R}^{\mathcal{W} \times \mathcal{H}}$, the mixed sample $(x_{(a,b)}, y_{(a,b)})$ can be generated as $x_{(a,b)} = H \odot x_a + (1 - H) \odot x_b$, and $y_{(a,b)} = \lambda y_a + (1 - \lambda) y_b$, where $\odot$ denotes element-wise product, $(x_a, y_a)$ and $(x_b, y_b)$ are sampled from a labeled dataset $L = \{(x_a, y_a)\}_{a=1}^{n_L}$. Note that superscripts denote the class index; subscripts indicate the type of data, *e.g.*, $x_{(a,b)}$ represents a mixed sample related to $x_a$ and $x_b$; $y^i$ is the value on $i$-th position. Since the mixup labels are obtained by linear interpolation, the standard CE loss weighted by $\lambda$, $\mathcal{L}_{MCE} = y_{(a,b)}^T \log \sigma(z_{(a,b)})$, is typically used as the optimization objective in the mixup training process, which can be rewritten as:

$$\mathcal{L}_{MCE} = -\sum_{i=1}^{C} \left( \lambda \mathbb{I}(y_a^i = 1) \log p_{(a,b)}^i + (1 - \lambda) \mathbb{I}(y_b^i = 1) \log p_{(a,b)}^i \right), \tag{1}$$

where $\mathbb{I}(\cdot) \in \{0, 1\}$ is an indicator function that values one if and only if the input condition holds; $p_{(a,b)}$. Noticeably, these two items of $\mathcal{L}_{MCE}$ are classifying $y_a$ and $y_b$ according to mixup coefficient $\lambda$. The effect of $\lambda$ weighting in standard cross-entropy loss is to smooth the decision space between two classes related in mixed samples. However, the condition for $\mathcal{L}_{MCE}$ to be optimal is $\frac{p_a}{p_b} = \frac{\lambda}{1-\lambda}$, which means $\mathcal{L}_{MCE}$ only focuses on regressing $\lambda$ value, but not encouraging the model to mine underlying statistics of mixed samples, *e.g.*, small $\lambda$ value suppresses the confidence of predictions even if the mixed features are evident. Therefore, the first challenge is to improve and design a general and efficient objective function for mixup training that maintains the smoothness of mixup and can simultaneously explore the key features related to the downstream task.

### 2.2 UNRELIABLE MIXUP AMPLIFIES CONFIRMATION BIAS

If we further consider semi-supervised learning (SSL), combining mixup and pseudo-labeling techniques can effectively improve performances with limited labeled data (Berthelot et al., 2019a). For each $u_a$ in an unlabeled dataset $U = \{(u_a, v_a)\}_{a=1}^{n_U}$, where $v_a$ its corresponding pseudo label, taking MixMatch (Berthelot et al., 2019b) as an example, pseudo-label $\hat{y}$ is generated by computing the average of the models predicted class distributions across all the $K$ basic augmentations (*e.g.,* image cropping, *etc.*) of $u_a$:

$$\hat{y}_a = \arg\max_C \left( \frac{1}{K} \sum_{k=1}^{K} M(y_a | \hat{u}_a; \theta) \right).$$

After obtaining the pseudo labels, we can use the same mixup process as before to generate virtual samples to augment the dataset. However, if the reliability of pseudo labels is not considered, the generated samples with false pseudo labels (FPLs) can even mislead the model and aggravate confirmation bias (Arazo et al., 2020), as shown in Figure 3 *left*. This becomes worse when mixup is employed on FPLs (supported by Sec. 4.3); either side with larger or smaller weights can further reinforce the wrong decision hyperplane and make it irreparable. Yet, if mixup is used only on labeled data, it will not

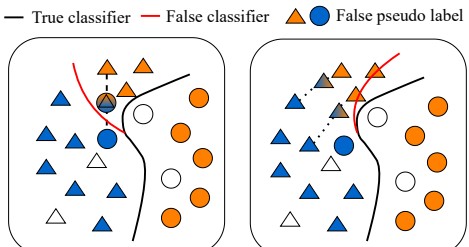

Figure 3: Left: unconstrained mixup misleads the model by generating bad samples. Right: the decision bound can be correctly guided by the DM.

be able to mine deeper into the potential information in a large amount of unlabeled data. Thus, we not only expect DM to improve the training efficiency in the supervised framework as Sec. 2.1 but also want to solve the confirmation bias under semi-supervision, as shown in Figure 3 *right*.

# 3 DATA-EFFICIENT MIXUP LEARNING

In data-efficient mixup learning, various scenarios derived from supervised learning (SL) will be considered, including transfer learning (TL) and SSL. The mentioned label mismatch and unreliable mixup are the two main challenges in these scenarios where mixup plays an important role. Both of them will be addressed by decoupled mixup simply and effectively.

## 3.1 DECOUPLED MIXUP

As stated in the preliminaries 2, the major problem of $\lambda$ weighted CE ($\mathcal{L}_{MCE}$) is that, while preserving the $\lambda$ consistency between $x_{(a,b)}$ and $y_{(a,b)}$ to smooth the predictions but neglect the potential information of mixed samples when the mixed label and sample are mismatching.

**Proposition 1.** *Assuming $x_{(a,b)}$ is generated from two different classes, minimizing $\mathcal{L}_{MCE}$ is equivalent to regress corresponding $\lambda$ in the gradient of $\mathcal{L}_{MCE}$:*

$$(\nabla_{z_{(a,b)}} \mathcal{L})^l = \begin{cases} -\lambda + \frac{\exp(z^l_{(a,b)})}{\sum_c \exp(z^c_{(a,b)})}, & l = i \\ -(1-\lambda) + \frac{\exp(z^l_{(a,b)})}{\sum_c \exp(z^c_{(a,b)})}, & l = j \\ \frac{\exp(z^l_{(a,b)})}{\sum_c \exp(z^c_{(a,b)})}, & l \neq i, j \end{cases} \tag{2}$$

where $l$ is the index of our interested class. As we can see, the predicted probability of $x_{(a,b)}$ is coupled with $\lambda$ in Equation 2. Such a design deliberately suppresses the prediction confidence for the class with a smaller $\lambda$, although it builds a linear correspondence according to $\lambda$. In other words, $\mathcal{L}_{MCE}$ does not encourage the model to focus on the class when its information is insufficient (label mismatch), but it could have been a good hard mixed sample. As a consequence, $\mathcal{L}_{MCE}$ makes many static mixup methods require a long training time to achieve the desired results (Verma et al., 2019; Yu et al., 2021). Based on previous analysis, a novel mixup objective, decoupled mixup, is raised to learn the comprehensive features of the mixed data adaptively and improve the mixup training efficiency. Before that, we first start with an original Softmax function. For mixed data points $z_{(a,b)}$ generated from a random pair in labeled dataset $L$, an encoded representation $z_{(a,b)} = f_\theta(x_{(a,b)})$ is generated by a feature extractor $f_\theta$. A mixup categorical probability of $i$-th class is attained:

$$\sigma(z_{(a,b)})^i = \frac{\exp(z^i_{(a,b)})}{\sum_c \exp(z^c_{(a,b)})}.$$

where the result of $\sigma(\cdot)$ is a vector of categorical probabilities. Empirically, a well-designed dynamic mixup policy that decouples the target object from the background effectively reduces label mismatch and thus improves performance. This motivated us to introduce decoupling to the objective function of mixup. For a mixed sample, we can achieve decoupling by omitting one of the mixed classes and thereby calculating the prediction probability distribution independently. Even if the mixed sample does not contain distinct class characteristics, the model will make relatively high confidence predictions without the competitor class, *e.g.,* another class in the mixup. That is, the model will be encouraged to treat that sample as a hard sample, expecting to mine the underlying feature. And vice versa, if the characteristics are obvious enough, the model's prediction is not limited by the $\lambda$ value. Therefore, we propose to remove the competitor class in the standard Softmax to achieve decoupled prediction. The score on $i$-th class is not affected by the $j$-th class:

$$\phi(z_{(a,b)})^{i,j} = \frac{\exp(z^i_{(a,b)})}{\exp(z^j_{(a,b)}) + \sum_{c \neq j} \exp(z^c_{(a,b)})}.$$

where the result of $\phi(\cdot)$ is a matrix of categorical probabilities. By removing the opponent class, the two desired effects are achieved simultaneously: (1) breaking the $\lambda$ restriction on prediction; (2) encouraging more confident prediction on the current class. Compared with Equation 2, the decoupled Softmax makes all items associated with $\lambda$ become -1, the derivation is given in the A.1.

**Proposition 2.** *With the decoupled Softmax defined above, decoupled mixup cross-entropy $\mathcal{L}_{DM(CE)}$ can boost the prediction confidence of the interested classes mutually and escape from the $\lambda$-constraint:*

$$\mathcal{L}_{DM(CE)} = -\sum_{i=1}^c \sum_{j=1}^c y^i_a y^j_b \left( \log\left(\frac{p^i_{(a,b)}}{1 - p^j_{(a,b)}}\right) + \log\left(\frac{p^j_{(a,b)}}{1 - p^i_{(a,b)}}\right) \right).$$

The proofs of Proposition 1 and 2 are given in Appendix. In this way, the scope and confidence of our target class probability are successfully decoupled from the "background". Further, according to the property of decoupled cross-entropy loss which only pays attention to the current classes, the original linear correspondence between the mixed sample and label could be corrupted. Therefore, to preserve the consistency, $\mathcal{L}_{MCE}$ loss is employed in the final Decoupled Mixup (DM) loss. The overall loss function of decoupled mixup can be formulated as follows:

$$\mathcal{L}_{DM} = -\big( \underbrace{y_{(a,b)}^T \log(\sigma(z_{(a,b)}))}_{\mathcal{L}_{MCE}} + \eta \underbrace{y_{[a,b]}^T \log(\phi(z_{(a,b)}))y_{[a,b]}}_{\mathcal{L}_{DM(CE)}} \big).$$

where $y_{(a,b)}$ indicates the mixed label while $y_{[a,b]}$ is two-hot label encoding, $\eta$ is a trade-off factor. Notice that $\eta$ is robust and can be set according to the character of mixup methods (see Sec. 4.4).

## 3.2 ASYMMETRICAL STRATEGY

Although $\mathcal{L}_{DM}$ can improve the learning efficiency in labeled data, how to effectively exploit the unlabeled data and reduce the confirmation bias is still a problem. Recall the confirmation bias problem of SSL: the performance of the student model is restricted by the teacher model when learning from inaccurate pseudo-labels. To strengthen the teacher to provide more accurate predictions, the unlabeled data with larger $\lambda$ can be used to mixup with the labeled data to form hard samples. Specifically, a larger $\lambda$ can make the unlabeled data obscure most of the information of the labeled data, but since the label is assured, the whole process of mixup can be treated as hard sample generation. With the addition of these hard samples, the teacher model will have better performance and thus give more reliable pseudo-labels. Formally, given the labeled and unlabeled datasets $L$ and $U$, AS builds reliable mixed samples between $L$ and $U$ with an asymmetric manner ($\lambda < 0.5$):

$$\hat{x}_{(a,b)} = \lambda x_a + (1-\lambda)u_b; \quad \hat{y}_{(a,b)} = \lambda y_a + (1-\lambda)v_b.$$

Due to the uncertainty of the pseudo-label, only the labeled part is retained in $\mathcal{L}_{DM(CE)}$:

$$\hat{\mathcal{L}}_{DM(CE)} = y_a^T \log\big(\phi(z_{(a,b)})\big)y_b,$$

where $y_a$ and $y_b$ are one-hot labels. AS could be regarded as a special case of DM that only decouples the hard samples with ground-truth labels. Replacing $\mathcal{L}_{DM(CE)}$ with $\hat{\mathcal{L}}_{DM(CE)}$ can leverage the hard samples in mixed samples between $L$ and $U$ while alleviating the problem of confirmation bias.

## 3.3 GENERALIZATION TO BINARY CROSS-ENTROPY

**Binary cross-entropy form of DM.** If we treat mixup training as a multi-label classification task (1-*vs*-all) using mixup binary cross-entropy (MBCE) loss (Wightman et al., 2021) ($\sigma(\cdot)$ denotes Sigmoid), then we can generate the decoupled loss for each class. Proposition 2 demonstrates the decoupled Softmax-based CE can mutually enhance the confidence of predictions for the interested classes and be free from $\lambda$ limitations. For MBCE, since it is not inherently bound to mutual interference between classes by Softmax, we need to preserve partial consistency and encourage more confident predictions to propose a decoupled mixup binary cross-entropy loss, DM(BCE).

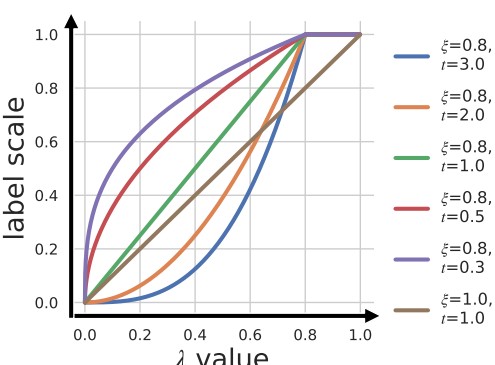

Figure 4: Rescaled label of different $\lambda$ value.

To this end, a rescaling function $r : \lambda, t, \xi \to \lambda'$ is designed to achieve this goal. The mixed label is enhanced by $r(\cdot)$: $y_{mix} = \lambda_a y_a + \lambda_b y_b$, where $\lambda_a$ and $\lambda_b$ are rescaled. The rescaling function is defined as follows:

$$r(\lambda, t, \xi) = \big(\frac{\lambda}{\xi}\big)^t, \quad 0 \le t, 0 \le \xi < 1, \tag{3}$$

where $\xi$ is the threshold, $t$ is an index to control the convexity. As shown in Figure 4, there are three situations: (a) when $\xi = 0$, $t = 0$, the rescaled label is always equal to 1, as two-hot encoding; (b) when $\xi = 1$, $t = 1$, $r(\cdot)$ is a linear function (vanilla mixup); (c) the rest curves demonstrate $t$ is the parameter that changes the concavity and $\xi$ is responsible for truncating.

**Empirical results.** In the case of interpolation-based mixup methods (*e.g.*, Mixup, ManifoldMix, *etc.*) that keep a linear relationship between the mixed label and sample, the decoupled mechanism can be introduced by only adjusting the threshold $t$. In the case of cutting-based mixing policies (*e.g.*, CutMix, PuzzleMix, *etc.*) where the mixed samples and labels have a square relationship (generally a convex function), we can approximate the convex function by adjusting $\xi$.

## 4 EXPERIMENTS

We adopt two types of top-1 classification accuracy (Acc) metrics (the mean of three trials): (i) the median top-1 Acc of the last 10 epochs (Sohn et al., 2020; Liu et al., 2022), and (ii) the best top-1 Acc in all checkpoints. We report the median top-1 Acc for image classification tasks with Mixup variants and the max top-1 Acc for SSL tasks. Popular ConvNets and Transformer-based architectures are used as backbone networks: ResNet variants including ResNet (He et al., 2016) (R), Wide-ResNet (WRN) (Zagoruyko & Komodakis, 2016), and ResNeXt-32x4d (RX) (Xie et al., 2017), Transformer-based architectures including DeiT (Touvron et al., 2021) and Swin Transformer (Swin) (Liu et al., 2021). Double horizontal line in the tables splits the *static* and *dynamic* methods.

### 4.1 IMAGE CLASSIFICATION BENCHMARKS

This subsection evaluates performance gains of DM on six image classification benchmarks, including CIFAR-10/100 (Krizhevsky et al., 2009), Tiny-ImageNet (Chrabaszcz et al., 2017), ImageNet-1k (Russakovsky et al., 2015), CUB-200-2011 (CUB) (Wah et al., 2011), FGVC-Aircraft (Aircraft) (Maji et al., 2013). There are mainly two types of mixup methods based on their mixing policies: *static* methods including Mixup (Zhang et al., 2017), CutMix (Yun et al., 2019), ManifoldMix (Verma et al., 2019), SaliencyMix (Uddin et al., 2020), FMix (Harris et al., 2020), and ResizeMix (Qin et al., 2020), and *dynamic* mixup methods including PuzzleMix (Kim et al., 2020), AutoMix (Liu et al., 2022), and SAMix (Li et al., 2021b). For a fair comparison, we use the optimal $\alpha$ in $\{0.1, 0.2, 0.5, 0.8, 1.0, 2.0\}$ for all mixup algorithms and follow original hyper-parameters in papers. We adopt the open-source codebase OpenMixup (Li et al., 2022) for most mixup methods. The detailed training recipes are provided in Appendix A.3.

Table 1: Top-1 Acc (%)↑ of small-scale image classification on CIFAR-100 and Tiny-ImageNet datasets based on ResNet variants.

| Datasets | CIFAR-100 | | | | | | Tiny-ImageNet | | | |
|---|---|---|---|---|---|---|---|---|---|---|
| | R-18 | | RX-50 | | WRN-28-8 | | R-18 | | RX-50 | |
| Methods | MCE | DM(CE) | MCE | DM(CE) | MCE | DM(CE) | MCE | DM(CE) | MCE | DM(CE) |
| Mixup | 79.12 | **80.44** | 82.10 | **82.96** | 82.82 | **83.51** | 63.86 | **65.07** | 66.36 | **67.70** |
| CutMix | 78.17 | **79.39** | 81.67 | **82.39** | 84.45 | **84.63** | 65.53 | **66.45** | 66.47 | **67.46** |
| ManifoldMix | 80.35 | **80.95** | 82.88 | **83.15** | 83.24 | **83.42** | 64.15 | **65.45** | 67.30 | **68.48** |
| FMix | 79.69 | **80.00** | 81.90 | **82.74** | 84.21 | **84.28** | 63.47 | **65.34** | 65.08 | **66.96** |
| ResizeMix | 80.01 | **80.06** | 81.82 | **82.96** | 84.87 | 84.72 | 63.74 | **64.33** | 65.87 | **68.56** |
| SaliencyMix | 79.12 | **79.28** | 81.53 | **82.61** | 84.35 | **84.41** | 64.60 | **66.56** | 66.55 | **67.52** |
| PuzzleMix | 81.13 | **81.34** | 82.85 | **82.97** | 85.02 | **85.25** | 65.81 | **66.52** | 67.83 | **68.04** |
| AutoMix | 82.04 | **82.32** | 83.64 | **83.94** | 85.18 | **85.38** | 67.33 | **68.18** | 70.72 | **71.56** |
| SAMix | 82.30 | **82.40** | 84.42 | **84.53** | 85.50 | **85.59** | 68.89 | **69.16** | 72.18 | **72.39** |

**Small-scale classification benchmarks.** Table 1 shows small-scale classification results, the proposed DM(CE) significantly improves MCE based on various mixup algorithms: it meets our expectation that DM(CE) brings more performance gains for most *static* mixup variants, *e.g.*, +1.32%/0.86%/0.69% on CIFAR-100 and +1.21%/1.34% on Tiny for Mixup, except for ResizeMix. It might be because ResizeMix tries to preserve the full target information of the two mixing classes. Meanwhile, DM(CE) still enhances *dynamic* mixup methods even though these algorithms have achieved high performance: DM(CE) brings 0.23%~0.36% on CIFAR-100 for the previous state-of-the-art PuzzleMix and brings 0.21%~0.27% on Tiny for the current best method SAMix.

**ImageNet and fine-grained classification benchmarks.** As shown in Table 2, 3, and 4, DM(CE) improves over MCE in all mixup algorithms on three training settings: around +0.4% for mixup methods based on ResNet variants using PyTorch-style and RSB A3 settings; around +0.5% and +0.2% for all methods based on DeiT-S and Swin-T using DeiT setting. Meanwhile, Table 3 shows that DM(BCE) noticeably boosts the original MBCE, *e.g.*, +1.21%/+0.68%/+0.79% for RSB/CutMix/ManifoldMix and +0.37%/+0.12% for AutoMix/SAMix. Notice that MBCE(two)

Table 2: Top-1 Acc (%)↑ of image classification on ImageNet-1k with ResNet variants using PyTorch-style training recipe.

| Methods | R-18 MCE | R-18 MD(CE) | R-34 MCE | R-34 MD(CE) | R-50 MCE | R-50 MD(CE) |
|---|---|---|---|---|---|---|
| Vanilla | 70.04 | - | 73.85 | - | 76.83 | - |
| Mixup | 69.98 | 70.20 | 73.97 | 74.26 | 77.12 | 77.41 |
| CutMix | 68.95 | 69.26 | 73.58 | 73.88 | 77.07 | 77.32 |
| ManifoldMix | 69.98 | 70.33 | 73.98 | 74.25 | 77.01 | 77.30 |
| FMix | 69.96 | 70.26 | 74.08 | 74.34 | 77.19 | 77.38 |
| ResizeMix | 69.50 | 69.90 | 73.88 | 74.00 | 77.42 | 77.65 |
| SaliencyMix | 69.16 | 69.57 | 73.56 | 73.92 | 77.14 | 77.42 |
| PuzzleMix | 70.12 | 70.32 | 74.26 | 74.51 | 77.54 | 77.71 |
| AutoMix | 70.51 | 70.64 | 74.52 | 74.77 | 77.91 | 78.15 |
| SAMix | 70.85 | 70.90 | 74.96 | 75.10 | 78.11 | 78.36 |

Table 3: Top-1 Acc (%)↑ of image classification on ImageNet-1k based on ResNet-50 using RSB A3 training recipe.

| Methods | MCE | MD(CE) | MBCE (one) | MBCE (two) | MD(BCE) (one) |
|---|---|---|---|---|---|
| RSB | 76.49 | 77.72 | 78.08 | 76.95 | 78.43 |
| Mixup | 76.01 | 76.69 | 77.66 | 77.42 | 78.28 |
| CutMix | 76.47 | 77.22 | 77.62 | 67.54 | 78.21 |
| ManifoldMix | 76.14 | 76.93 | 77.78 | 67.78 | 78.20 |
| FMix | 76.09 | 76.87 | 77.76 | 73.44 | 78.11 |
| ResizeMix | 76.90 | 77.21 | 77.85 | 77.30 | 78.32 |
| SaliencyMix | 76.85 | 77.25 | 77.93 | 72.74 | 78.24 |
| PuzzleMix | 77.27 | 77.60 | 78.02 | 77.19 | 78.15 |
| AutoMix | 77.45 | 77.82 | 78.33 | 77.46 | 78.62 |
| SAMix | 78.33 | 78.45 | 78.64 | 77.58 | 78.75 |

Table 4: Top-1 Acc (%)↑ of classification on ImageNet-1k with ViTs.

| Methods | DeiT-S MCE | DeiT-S MD(CE) | Swin-T MCE | Swin-T MD(CE) |
|---|---|---|---|---|
| DeiT | 79.80 | 80.37 | 81.28 | 81.49 |
| Mixup | 79.65 | 80.04 | 80.71 | 80.97 |
| CutMix | 79.78 | 80.20 | 80.83 | 81.05 |
| FMix | 79.41 | 79.89 | 80.37 | 80.54 |
| ResizeMix | 79.93 | 80.03 | 80.94 | 81.01 |
| SaliencyMix | 79.32 | 79.86 | 80.68 | 80.83 |
| PuzzleMix | 79.84 | 80.25 | 81.03 | 81.16 |
| AutoMix | 80.78 | 80.91 | 81.80 | 81.92 |
| SAMix | 80.94 | 81.12 | 81.87 | 81.97 |

Table 5: Top-1 Acc (%)↑ of fine-grained image classification on CUB-200 and FGVC-Aircrafts datasets with ResNet variants.

| Datasets | CUB-200 | | | | FGVC-Aircrafts | | | |
|---|---|---|---|---|---|---|---|---|
| | R-18 | | RX-50 | | R-18 | | RX-50 | |
| Methods | MCE | DM(CE) | MCE | DM(CE) | MCE | DM(CE) | MCE | DM(CE) |
| Mixup | 78.39 | 79.90 | 84.58 | 85.04 | 79.52 | 82.66 | 85.18 | 86.68 |
| CutMix | 78.40 | 78.76 | 85.68 | 85.97 | 78.84 | 81.64 | 84.55 | 85.75 |
| ManifoldMix | 79.76 | 79.92 | 86.38 | 86.42 | 80.68 | 82.57 | 86.60 | 86.92 |
| FMix | 77.28 | 80.10 | 84.06 | 84.85 | 79.36 | 80.44 | 84.85 | 85.04 |
| ResizeMix | 78.50 | 79.58 | 84.77 | 84.92 | 78.10 | 79.54 | 84.08 | 84.51 |
| SaliencyMix | 77.95 | 78.28 | 83.29 | 84.51 | 80.02 | 81.31 | 84.31 | 85.07 |
| AutoMix | 79.87 | 81.08 | 86.56 | 86.74 | 81.37 | 82.18 | 86.69 | 86.82 |
| SAMix | 81.11 | 81.27 | 86.83 | 86.95 | 82.15 | 83.68 | 86.80 | 87.22 |

denotes using two-hot encoding for corresponding mixing classes, which yield worse performance than MBCE, and DM(BCE) adjusts the labels for the mixing classes by Equation 3. It verifies the necessity of MD(BCE) in the case of using MBCE. Table 5 shows that DM(CE) noticeably boosts the original MCE for eight popular mixup variants, especially bringing 0.53%~3.14% gains on Aircraft based on ResNet-18.

## 4.2 TRANSFER LEARNING BENCHMARKS

Following the transfer learning (TL) benchmarks (You et al., 2020), we perform TL experiments on CUB, Aircraft, and Stanford-Cars (Krause et al., 2013) (Cars). Besides the vanilla Fine-Tuning baseline, we compare current state-of-the-art TL methods, including BSS (Xinyang et al., 2019), Co-Tuning (You et al., 2020), and Self-Tuning (Ximei et al., 2021). For a fair comparison, we use the same hyper-parameters and augmentations as Self-Tuning: all methods are initialized by PyTorch pre-trained models on ImageNet-1k and trained 27k steps in total by SGD optimizer with the basic learning rate of 0.001, the momentum of 0.9, and the weight decay of 0.0005. In Table 6, we adopt DM(CE) and AS for Fine-Tuning, Co-Tuning, and Self-Tuning using Mixup. DM(CE) and AS steadily improve Mixup and the baselines by large margins, e.g., +4.62%~9.19% for 15% labels, +2.02%~5.67% for 30% labels, and +2.09%~3.15% for 50% labels on Cars. This outstanding improvement implies that generating mixed samples efficiently is essential for data-efficient training in data-limited scenarios. A similar performance will be presented as well in the next SSL setting.

## 4.3 SEMI-SUPERVISED LEARNING BENCHMARKS

Following (Sohn et al., 2020; Zhang et al., 2021), we adopt the most commonly-used CIFAR-10/100 datasets among the famous SSL benchmarks based on WRN-28-2 and WRN-28-8. We mainly evaluate

Table 6: Top-1 Acc (%)↑ of transfer learning on various TL benchmarks using only 15%, 30% and 50% labels based on ResNet-50.

| Methods | CUB-200 15% | CUB-200 30% | CUB-200 50% | FGVC-Aircraft 15% | FGVC-Aircraft 30% | FGVC-Aircraft 50% | Stanford-Cars 15% | Stanford-Cars 30% | Stanford-Cars 50% |
|---|---|---|---|---|---|---|---|---|---|
| Fine-Tuning | 45.25±0.12 | 59.68±0.21 | 70.12±0.29 | 39.57±0.20 | 57.46±0.12 | 67.93±0.28 | 36.77±0.12 | 60.63±0.18 | 75.10±0.21 |
| +DM | 50.04±0.17 | 61.39±0.24 | 71.87±0.23 | 43.15±0.22 | 61.02±0.15 | 70.38±0.18 | 41.30±0.16 | 62.65±0.21 | 77.19±0.19 |
| BSS | 47.74±0.23 | 63.38±0.29 | 72.56±0.17 | 40.41±0.12 | 59.23±0.31 | 69.19±0.13 | 40.57±0.12 | 64.13±0.18 | 76.78±0.21 |
| Co-Tuning | 52.58±0.53 | 66.47±0.17 | 74.64±0.36 | 44.09±0.67 | 61.65±0.32 | 72.73±0.08 | 46.02±0.18 | 69.09±0.10 | 80.66±0.25 |
| +DM | 54.96±0.65 | 68.25±0.21 | 75.72±0.37 | 49.27±0.83 | 65.60±0.14 | 74.89±0.17 | 51.78±0.34 | 74.15±0.29 | 83.02±0.26 |
| Self-Tuning | 64.17±0.47 | 75.13±0.35 | 80.22±0.36 | 64.11±0.32 | 76.03±0.25 | 81.22±0.29 | 72.50±0.45 | 83.58±0.28 | 88.11±0.29 |
| +Mixup | 62.38±0.32 | 74.65±0.24 | 81.46±0.27 | 59.38±0.31 | 74.65±0.26 | 81.46±0.27 | 70.31±0.27 | 83.63±0.23 | 88.66±0.21 |
| +DM | 73.06±0.38 | 79.50±0.35 | 82.64±0.24 | 67.57±0.27 | 80.71±0.25 | 84.82±0.26 | 81.69±0.23 | 89.22±0.21 | 91.26±0.19 |

Table 7: Top-1 Acc (%)↑ of semi-supervised learning on CIFAR-10 (using 250 and 4000 labels) and CIFAR-100 (using 400, 2500, and 10000 labels) based on WRN-28-2 and WRN-28-8, respectively. Notice that DM denotes using DM(CE) and AS, Con denotes various unsupervised consistency losses, Rot denotes the rotation loss in ReMixMatch, and CPL denotes the curriculum labeling in FlexMatch.

| | | CIFAR-10 | | CIFAR-100 | | |
|---|---|---|---|---|---|---|
| Methods | Losses | 250 | 4000 | 400 | 2500 | 10000 |
| Pseudo-Labeling | CE | $53.51_{\pm2.20}$ | $84.92_{\pm0.19}$ | $12.55_{\pm0.85}$ | $42.26_{\pm0.28}$ | $63.45_{\pm0.24}$ |
| MixMatch | CE+Con | $86.37_{\pm0.59}$ | $93.34_{\pm0.26}$ | $32.41_{\pm0.66}$ | $60.24_{\pm0.48}$ | $72.22_{\pm0.29}$ |
| ReMixMatch | CE+Con+Rot | $93.70_{\pm0.05}$ | $95.16_{\pm0.01}$ | $57.15_{\pm1.05}$ | $73.87_{\pm0.35}$ | $79.08_{\pm0.27}$ |
| **MixMatch+DM** | CE+Con+DM | $89.16_{\pm0.71}$ | $95.15_{\pm0.68}$ | $35.72_{\pm0.53}$ | $62.51_{\pm0.37}$ | $74.70_{\pm0.28}$ |
| UDA | CE+Con | $94.84_{\pm0.06}$ | $95.71_{\pm0.07}$ | $53.61_{\pm1.59}$ | $72.27_{\pm0.21}$ | $77.51_{\pm0.23}$ |
| FixMatch | CE+Con | $95.14_{\pm0.05}$ | $95.79_{\pm0.08}$ | $53.58_{\pm0.82}$ | $71.97_{\pm0.16}$ | $77.80_{\pm0.12}$ |
| FlexMatch | CE+Con+CPL | $95.02_{\pm0.09}$ | $95.81_{\pm0.01}$ | $\mathbf{60.06_{\pm1.62}}$ | $73.51_{\pm0.20}$ | $78.10_{\pm0.15}$ |
| FixMatch+Mixup | CE+Con+MCE | $95.05_{\pm0.23}$ | $95.83_{\pm0.19}$ | $50.61_{\pm0.73}$ | $72.16_{\pm0.18}$ | $78.75_{\pm0.14}$ |
| **FixMatch+DM** | CE+Con+DM | $\mathbf{95.23_{\pm0.09}}$ | $\mathbf{95.87_{\pm0.11}}$ | $59.75_{\pm0.95}$ | $\mathbf{74.12_{\pm0.23}}$ | $\mathbf{79.58_{\pm0.17}}$ |

the proposed DM on popular SSL methods MixMatch (Berthelot et al., 2019b) and FixMatch (Sohn et al., 2020), and compare with Pesudo-Labeling (Lee et al., 2013), ReMixMatch (Berthelot et al., 2019a), UDA (Xie et al., 2019), and FlexMatch (Zhang et al., 2021). For a fair comparison, we use the same hyperparameters and training settings as the original papers and adopt the open-source codebase TorchSSL (Zhang et al., 2021) for all methods. Concretely, we use an SGD optimizer with a basic learning rate of $lr = 0.03$ adjusted by Cosine Scheduler, the total $2^{20}$ steps, the batch size of 64 for labelled data, and the confidence threshold $\tau = 0.95$. Table 7 shows that adding DM(CE) and AS significantly improves MixMatch and FixMatch: DM(CE) brings 1.81∼2.89% gains on CIFAR-10 and 1.27∼3.31% gains on CIFAR-100 over MixMatch while bringing 1.78∼4.17% gains on CIFAR-100 over FixMatch. Meanwhile, we find that directly applying mixup augmentations to FixMatch brings limited improvements, while FixMatch+DM achieves the best performance in most cases on CIFAR-10/100 datasets. Appendix A.5 provides further studies with limited labelled data. Therefore, mixup with DM can achieve data-efficient training in semi-supervised scenarios.

## 4.4 ABLATION STUDY

Since we have demonstrated the effectiveness of DM in the above four subsections and Figure 6, we verify whether DM is robust to hyper-parameters (see full hyper-parameters in Appendix A.4) and study the effectiveness of AS in SSL tasks:

(1) The only hyper-parameter $\eta$ in DM(CE) and DM(BCE) can be set according to the types of mixup methods. We grid search $\eta$ in $\{0.01, 0.1, 0.5, 1, 2\}$ on ImageNet-1k. As shown in Figure 5 *left*, the *static* (Mixup and CutMix) and the *dynamic* methods (PuzzleMix and AutoMix) prefer $\eta = 0.1$ and $\eta = 1$, respectively, which might be because the *dynamic* variants generate more discriminative and reliable mixed samples than the *static* methods.

(2) Hyper-parameters $\xi$ and $t$ in DM(BCE) can also be determined by the characters of mixup policies. We grid search $\xi \in \{1, 0.9, 0.8, 0.7\}$ and $t \in \{2, 1, 0.5, 0.3\}$. Figure 5 *middle* and *right* show that cutting-based methods (CutMix and AutoMix) prefer $\xi = 0.8$ and $t = 1$, while the interpolation-based policies (Mixup and ManifoldMix) use $\xi = 1.0$ and $t = 0.5$.

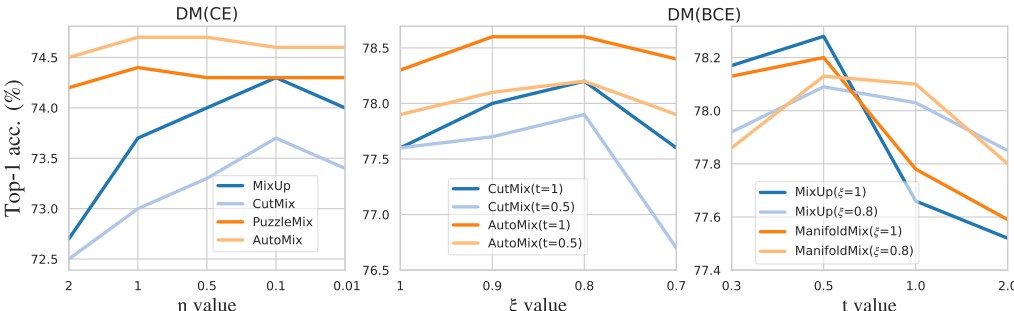

Figure 5: Ablation of hyper-parameters on ImageNet-1k based on ResNet-34. **Left**: analyzing the balancing weight $\eta$ in DM(CE); **Middle**: analyzing $\xi$ in DM(BCE) when $t$ is fixed to 1 and 0.5; **Right**: analyzing $t$ in DM(BCE) when $\xi$ is fixed to 1 and 0.8.

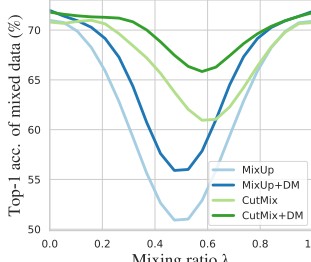

Figure 6: Top-1 Acc of mixed samples on ImageNet-1k validation.

Table 8: Ablation of the proposed asymmetric strategy (AS) and DM(CE) of semi-supervised transfer learning on CUB-200 based on ResNet-18.

| Methods | 15% | 30% | 50% |
|---|---|---|---|
| Self-Tuning | 57.82 | 69.12 | 73.59 |
| +MCE | 63.36 | 72.81 | 75.73 |
| +MCE+AS($\lambda \geq 0.5$) | 59.04 | 69.67 | 74.89 |
| +MCE+AS($\lambda \leq 0.5$) | 62.97 | 72.46 | 75.40 |
| +DM(CE)+AS($\lambda \leq 0.5$) | **66.17** | **74.25** | **77.68** |

(3) Table 8 shows the superior of AS($\lambda \leq 0.5$) in comparison to MCE and AS($\lambda \geq 0.5$), while using DM(CE) and AS($\lambda \leq 0.5$) further improves MCE.

(4) The experiments of different sizes of training data are performed to verify the data-efficientness of DM. We can observe that decoupled mixup improves by around 2% accuracy without any computational overhead. The detailed results are shown in Appendix **??**.

## 5 RELATED WORK

**Mixup.** As data-dependent augmentation techniques, mixup methods generate new samples by mixing two or more samples and corresponding labels with well-designed mixing policies (Zhang et al., 2017; Yun et al., 2019; Verma et al., 2019; Kim et al., 2020; 2021; Liu et al., 2022; Li et al., 2021b). The pioneering mixing method is Mixup (Zhang et al., 2017), whose mixed samples are generated by linear interpolation between pairs of samples. After that, cut-based methods are proposed to improve the mixup for localizing important features, especially in the vision field. In other works, authors explore using nonlinear or optimizable interpolation mixup policies, such as PuzzleMix (Kim et al., 2020), Co-Mixup (Kim et al., 2021), AutoMix (Liu et al., 2022), and SAMix (Li et al., 2021b). Moreover, mixup methods extend to more than two elements (Kim et al., 2021; Dabouei et al., 2021), and are utilized in contrastive learning to learn discriminative visual representation (Kalantidis et al., 2020; Lee et al., 2021; Shen et al., 2021; Li et al., 2021b).

**Semi-supervised Learning and Transfer Learning.** MixMatch (Berthelot et al., 2019b) and ReMixMatch (Berthelot et al., 2019a) apply mixup on labelled and unlabeled data to enhance the diversity of the dataset. More accurate pseudo-labelling relies on data augmentation techniques to introduce consistency regularization, *e.g.*, UDA (Xie et al., 2019) and FixMatch (Sohn et al., 2020) employ weak and strong augmentations to improve the consistency. Furthermore, CoMatch (Li et al., 2021a) unifies consistency regularization, entropy minimization, and graph-based contrastive learning to mitigate confirmation bias. FlexMatch (Zhang et al., 2021) improves FixMatch by applying curriculum learning for dynamically updating confidence threshold class-wisely. Fine-tuning a pre-trained model on labelled datasets is a widely adopted form of transfer learning (TL) in various applications. Previously, (Donahue et al., 2014; Oquab et al., 2014) show that transferring pre-trained AlexNet features to downstream tasks outperforms hand-crafted features. More recently, Self-Tuning (Ximei et al., 2021) introduced contrastive learning into TL to tackle confirmation bias and model shift issues in a one-stage framework.

## 6 CONCLUSION AND LIMITATIONS

In this paper, we introduce Decoupled Mixup (DM), a new objective function for considering both smoothing the decision boundaries and mining discriminative features. The proposed DM helps early *static* mixup methods (*e.g.,* MixUp and CutMix) achieve a comparable or better performance than the computationally expensive *dynamic* mixup policies. Most impotently, DM raises a question worthy of researching: *is it necessary to design very complex mixup policies to achieve expected results?* Based on our analysis and experiments, we argue that the loss function of mixup training is a new question worthy of studying. However, the introduction of additional hyperparameters may take users some extra time to check whether the default hyperparameters are reliable on datasets other than images. This also leads to the core question of the next step in the development of this work: how to design a more elegant and adaptive mixup training objective by leveraging label-mismatched samples?

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

# A APPENDIX

In this Appendix, we provide proofs of the proposition 1 (§A.1) and proposition 2 (§A.2) and implementation details (§A.3).

## A.1 PROOF OF PROPOSITION 1

**Proposition 1.** Assuming $x_{(a,b)}$ is generated from two different classes, minimizing $\mathcal{L}_{MCE}$ is equivalent to regress corresponding $\lambda$ in the gradient of $\mathcal{L}_{MCE}$:

$$(\nabla_{z_{(a,b)}}\mathcal{L}_{MCE})^i = \begin{cases} -\lambda + \frac{\exp(z^i_{(a,b)})}{\sum_c \exp(z^c_{(a,b)})}, & l = i \\ -(1-\lambda) + \frac{\exp(z^j_{(a,b)})}{\sum_c \exp(z^c_{(a,b)})}, & l = j \\ \frac{\exp(z^i_{(a,b)})}{\sum_c \exp(z^c_{(a,b)})}, & l \neq i,j \end{cases} \tag{4}$$

*Proof.* For the mixed sample $(x_{(a,b)}, y_{(a,b)})$, $z_{(a,b)}$ is derived from a feature extractor $f_\theta$ (i.e $z_{(a,b)} = f_\theta(x_{(a,b)})$). According to the definition of the mixup cross-entropy loss $\mathcal{L}_{MCE}$, we have:

$$\begin{aligned} \left(\nabla_{z(a,b)}\mathcal{L}_{MCE}\right)^l &= \frac{\partial \mathcal{L}_{MCE}}{\partial z^l_{(a,b)}} \\ &= -\frac{\partial}{\partial z^l_{(a,b)}}\left(y^T_{(a,b)}\log\left(\sigma(z_{(a,b)})\right)\right) \\ &= -\sum_{i=1}^C \left(y^i_{(a,b)}\frac{\partial}{\partial z^l_{(a,b)}}\left(\log\left(\frac{\exp(z^i_{(a,b)})}{\sum_{j=1}^C \exp(z^j_{(a,b)})}\right)\right)\right) \\ &= -\sum_{i=1}^C \left(y^i_{(a,b)}\frac{\sum_{j=1}^C \exp(z^j_{(a,b)})}{\exp(z^i_{(a,b)})}\frac{\partial}{\partial z^l_{(a,b)}}\left(\frac{\exp(z^i_{(a,b)})}{\sum_{j=1}^C \exp(z^j_{(a,b)})}\right)\right) \\ &= -\sum_{i=1}^C \left(y^i_{(a,b)}\left(\delta^l_i - \frac{\exp(z^l_{(a,b)})}{\sum_{j=1}^C \exp(z^j_{(a,b)})}\right)\right) \\ &= \frac{\exp(z^l_{(a,b)})}{\sum_{j=1}^C \exp(z^j_{(a,b)})} - y^l_{(a,b)} \end{aligned}$$

## A.2 PROOF OF PROPOSITION 2

**Proposition 2.** With the decoupled Softmax defined above, decoupled mixup cross-entropy $\mathcal{L}_{DM(CE)}$ can boost the prediction confidence of the interested classes mutually and escape from the $\lambda$-constraint:

$$\mathcal{L}_{DM(CE)} = \sum_{i=1}^{c} \sum_{j=1}^{c} y_a^i y_b^j \left( \log\big(\frac{p_{(a,b)}^i}{1 - p_{(a,b)}^j}\big) + \log\big(\frac{p_{(a,b)}^j}{1 - p_{(a,b)}^i}\big) \right).$$

*Proof.* For the mixed sample $(x_{(a,b)}, y_{(a,b)})$, $z_{(a,b)}$ is derived from a feature extractor $f_\theta$ (i.e $z_{(a,b)=f_\theta(x_{(a,b)})}$). According to the definition of the mixup cross-entropy loss $\mathcal{L}_{DM(CE)}$, we have:

$$\mathcal{L}_{DM(CE)} = y_{[a,b]}^T \log\big(H(Z_{(a,b)})\big)y_{[a,b]} \triangleq y_a^T \log\big(H(Z_{(a,b)})\big)y_b + y_b^T \log\big(H(Z_{(a,b)})\big)y_a$$

$$= \sum_{i,j=1}^{C} y_a^i \log\big(\frac{\exp(z_{(a,b)}^i)}{\sum_{k\neq j}^{C} \exp(z_{(a,b)}^j)}\big)y_b^j + \sum_{i,j=1}^{C} y_a^j \log\big(\frac{\exp(z_{(a,b)}^j)}{\sum_{k\neq i}^{C} \exp(z_{(a,b)}^i)}\big)y_b^i$$

$$= \sum_{i,j=1}^{C} y_a^i y_b^j \big( \log\big(\frac{\exp(z_{(a,b)}^i)}{\sum_{k\neq j}^{C} \exp(z_{(a,b)}^j)}\big) + \log\big(\frac{\exp(z_{(a,b)}^j)}{\sum_{k\neq i}^{C} \exp(z_{(a,b)}^i)}\big) \big)$$

$$= \sum_{i,j=1}^{C} y_a^i y_b^j \big( \log\big(\frac{\frac{\exp(z_{(a,b)}^i)}{\sum_{k=1}^{C} \exp(z_{(a,b)}^k)}}{\frac{\sum_{k\neq j}^{C} \exp(z_{(a,b)}^j)}{\sum_{k=1}^{C} \exp(z_{(a,b)}^k)}}\big) + \log\big(\frac{\frac{\exp(z_{(a,b)}^j)}{\sum_{k=1}^{C} \exp(z_{(a,b)}^k)}}{\frac{\sum_{k\neq i}^{C} \exp(z_{(a,b)}^i)}{\sum_{k=1}^{C} \exp(z_{(a,b)}^k)}}\big) \big)$$

$$= \sum_{i,j=1}^{C} y_a^i y_b^j \big( \log\big(\frac{p_{(a,b)}^i}{1 - p_{(a,b)}^j}\big) + \log\big(\frac{p_{(a,b)}^j}{1 - p_{(a,b)}^i}\big) \big)$$

where $p_{(a,b)} = \sigma(z_{(a,b)})$

## A.3 IMPLEMENTATION DETAILS

**Dataset.** We briefly introduce used image datasets. (1) Small scale classification benchmarks: CIFAR-10/100 (Krizhevsky et al., 2009) contains 50,000 training images and 10,000 test images in 32×32 resolutions, with 10 and 100 classes settings. Tiny-ImageNet (Chrabaszcz et al., 2017) is a rescaled version of ImageNet-1k, which has 10,000 training images and 10,000 validation images of 200 classes in 64×64 resolutions. (2) Large scale classification benchmarks: ImageNet-1k (Krizhevsky et al., 2012) contrains 1,281,167 training images and 50,000 validation images of 1000 classes. (3) Small-scale fine-grained classification scenarios: CUB-200-2011 (Wah et al., 2011) contains 11,788 images from 200 wild bird species for fine-grained classification. FGVC-Aircraft (Maji et al., 2013) contains 10,000 images of 100 classes of aircraft. Standford-Cars (Krause et al., 2013).

**Training settings on ImageNet-1k.** Tab. A1 shows the full training settings of PyTorch, DeiT, and RSB A3 on ImageNet-1k. Following (Yun et al., 2019), we replace the step learning rate decay with Cosine Scheduler (Loshchilov & Hutter, 2016) for better performances.

**Hyper-parameter settings.** We follow the basic hyper-parameter settings (*e.g.,* $\alpha$) for mixup variants in OpenMixup (Li et al., 2022), where we reproduce most comparison methods. Notice that *static* methods denote Mixup (Zhang et al., 2017), CutMix (Yun et al., 2019), ManifoldMix (Verma et al., 2019), SaliencyMix (Uddin et al., 2020), FMix (Harris et al., 2020), ResizeMix (Qin et al., 2020), and *dynamic* methods denote PuzzleMix (Kim et al., 2020), AutoMix (Liu et al., 2022), and SAMix (Li et al., 2021b)). Similarly, *interpolation-based* methods denote Mixup and ManifoldMix while *cutting-based* methods denote the rest mixup variants mentioned above. We set the hyper-parameters of DM(CE) as follows: For CIFAR-100 and ImageNet-1k, *static* methods use $\eta = 0.1$, and *dynamic* methods use $\eta = 1$. For Tiny-ImageNet and fine-grained datasets, *static* methods use $\eta = 1$ based on ResNet-18 while $\eta = 0.1$ based on ResNeXt-50; *dynamic* methods use $\eta = 1$. As for the hyper-parameters of DM(BCE) on ImageNet-1k, *cutting-based* methods use $t = 1$ and $\xi = 0.8$, while *interpolation-based* methods use $t = 0.5$ and $\xi = 1$. Note that we use $\alpha = 0.2$ and $\alpha = 2$ for the *static* and *dynamic* methods when using the proposed DM.

Table A1: Ingredients and hyper-parameters used for ImageNet-1k training settings.

| Procedure | PyTorch (Yun et al., 2019) | DeiT (Touvron et al., 2021) | RSB A3 (Wightman et al., 2021) |
|---|---|---|---|
| Train Res | $224^2$ | $224^2$ | $224^2$ |
| Test Res | $224^2$ | $224^2$ | $224^2$ |
| Test crop ratio | 0.875 | 0.875 | 0.95 |
| Epochs | 100/300 | 300 | 100 |
| Batch size | 256 | 1024 | 2048 |
| Optimizer | SGD | AdamW | LAMB |
| LR | 0.1 | $1 \times 10^{-3}$ | $8 \times 10^{-3}$ |
| LR decay | cosine | cosine | cosine |
| Weight decay | $10^{-4}$ | 0.05 | 0.02 |
| optimizer momentum | 0.9 | $\beta_1, \beta_2 = 0.9, 0.999$ | ✗ |
| Warmup epochs | ✗ | 5 | 5 |
| Label smoothing $\epsilon$ | ✗ | 0.1 | ✗ |
| Dropout | ✗ | ✗ | ✗ |
| Stoch. Depth | ✗ | 0.1 | 0.05 |
| Repeated Aug | ✗ | ✓ | ✓ |
| Gradient Clip. | ✗ | 1.0 | ✗ |
| H. flip | ✓ | ✓ | ✓ |
| RRC | ✓ | ✓ | ✓ |
| Rand Augment | ✗ | 9/0.5 | 6/0.5 |
| Auto Augment | ✗ | ✗ | ✗ |
| Mixup alpha | ✗ | 0.8 | 0.1 |
| Cutmix alpha | ✗ | 1.0 | 1.0 |
| Erasing prob. | ✗ | 0.25 | ✗ |
| ColorJitter | ✗ | ✗ | ✗ |
| EMA | ✗ | 0.99996 | ✗ |
| CE loss | ✓ | ✓ | ✗ |
| BCE loss | ✗ | ✗ | ✓ |

## A.4 Sensitivity Analysis

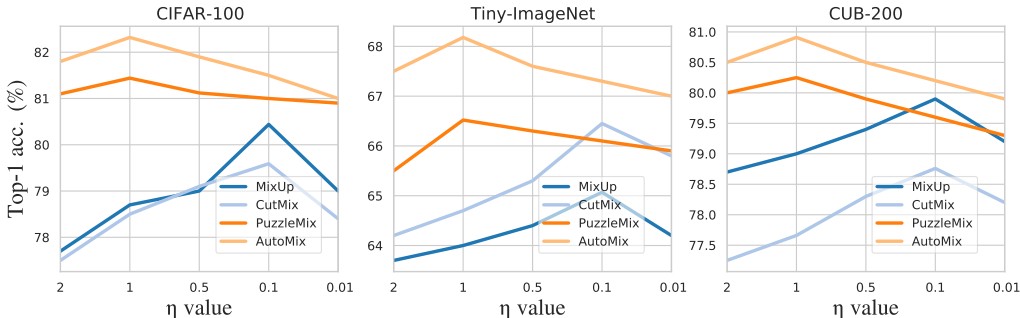

Figure A1: Sensitivity analysis of hyper-parameters on different datasets based on ResNet-18.

In order to verify the robustness of hyper-parameter $\eta$, extra experiments are conducted on various datasets: CIFAR-100, Tiny-ImageNet, and CUB-200. Figure A1 shows the results that are consistent with our ablation study in Section 4.4. The dynamic mixup methods prefer the large value of $\eta$ (1.0), while the static ones are more like a small value (0.1). The main reason for this is that dynamic methods generate mixed samples where label mismatch is relatively rare, relying on larger weights to achieve better results, while the opposite is true in static methods.

## A.5 Analysis of data-efficient mixup

To further DM whether data-efficient mixup training can be truly achieved, we conducted supervised experiments on CIFAR-100 with different sizes of training data. 15%, 30% and 50% of the CIFAR-100 data are randomly selected as training data, and the test data are unchanged. decoupled mixup uses DM(CE) as the loss function by default. From Table A2, we can see that DM improves the performance consistently without any computational overhead. Especially when using only 15% of the data, DM can improve accuracy by 2%. Therefore, combined with the experimental results of semi-supervised learning in Sec. 4.3 and Sec. 4.2, we can say that mixup training with DM is more data-efficient with limited data.

Table A2: Top-1 Acc (%)↑ of image classification on CIFAR-100 dataset with ResNet-18 using only 15%, 30%, and 50% labeled training sets.

| Methods | 15% | | 30% | | 50% | |
| --- | --- | --- | --- | --- | --- | --- |
| | MCE | DM(CE) | MCE | DM(CE) | MCE | DM(CE) |
| Vanilla | 42.48 | - | 56.41 | - | 64.32 | - |
| Mixup | 42.23 | **44.39** | 55.61 | **56.78** | 64.55 | **65.92** |
| CutMix | 43.81 | **44.85** | 55.99 | **57.14** | 64.38 | **65.87** |
| SaliencyMix | 42.95 | **44.01** | 55.42 | **56.51** | 64.56 | **66.10** |
| PuzzleMix | 42.67 | **43.87** | 56.19 | **57.36** | 64.74 | **66.26** |

