# OpenReview forum: "Decoupled Mixup for Data-efficient Learning"
_ICLR.cc/2023/Conference — Submitted to ICLR 2023_

### Official Review · Reviewer_A9VU · 2022-10-19

**Confidence:** 5
**Correctness:** 2
**Technical Novelty And Significance:** 2
**Empirical Novelty And Significance:** 3
**Recommendation:** 5

**Clarity, Quality, Novelty And Reproducibility:**

I think the clarity of this paper is just okay, and this paper has room for improvement in terms of clarity. Especially, I think this paper has a very weak connection between motivation and the method (Please see weakness for details). I also left some comments related to the formatting. I feel some statements are not supported well (e.g., "small lambda value suppress the confidence of predictions even if the mixed features are evident" => why and how will this be harmful to the training?). I suggest revising the paper to focus on the effect of decoupling rather than the label mismatch problem.

The quality in terms of the experiments looks excellent. However, I think the technical contribution and novelty of this paper (for solving the label mismatch problem as motivation) are not very significant, considering that the proposed method looks invariant to the label mismatch problem.

I think this paper will be reproducible relatively easily, considering that the base settings of this paper are widely-used ones (e.g., 300 epoch ImageNet-1k, DeiT training).

**Strength And Weaknesses:**

## Strength

- Because the proposed method is invariant to the choice of the mixing strategy, the proposed method is easy to be adopted to any MSDA method.
- The strongest strength of this paper is the expensive experimental results and consistent improvements.
    - The extensive study shows that the proposed method is effective to the existing MSDA methods. Especially, this paper studies the effect of the proposed objective function on various benchmarks with state-of-the-art optimization techniques. For example, this paper explores not only ImageNet training with 300 epochs and cosine lr scheduling (a standard one) but also ResNet Strike Back (RSB)-ish advanced optimization settings or DeiT optimization settings.
    - In my opinion, the ImageNet-1k experimental results (Table 2, 3, 4) will be helpful to researchers who work with ImageNet-1k benchmarks; as far as the reviewer knows, there are not many works that compare the existing state-of-the-art MSDA methods (Mixup, CutMix, ManifoldMixup, FMix, ResizeMix, SaliencyMix, PuzzleMix, AutoMix, SAMix) with the most advanced optimization settings, such as RSB or DeiT settings. E.g., considering the heavy computational resources of "dynamic MSDA" methods (SaliencyMix, PuzzleMix, AutoMix, SAMix), practitioners can choose ResizeMix + DM loss.
    - This paper also explores transfer learning and semi-supervised learning scenario in state-of-the-art settings.


## Weakness

### Weak connection between label mismatching problem and the proposed method

I cannot find any connection between the "label mismatching problem" and the "decoupled Mixup loss". Assuming we use a static Mixup method (e.g., Mixup), even if we use the DM loss, we still suffer from the label mismatching problem because the loss function still contains the original loss function. Recall that the label mismatching problem means that there exists a mismatch between a mixed sample and a mixed label; it means that if we want to solve this problem, we have to make a mixed label depending on a mixed sample. However, the mixed samples and the mixed labels by the proposed method are invariant; it will still suffer from the label mismatching problem if the base method suffers from the problem.

Although this paper has a meaningful empirical contribution, in terms of academic publication, I think this paper should need more verifications for the statements and the motivation. I cannot find any connection between the proposed method and the motivation.

Also, as a minor comment, I cannot find any connection between "data-efficient" (the title) and the proposed method. I think there need to be more statements for why this method is data-efficient compared to the original mixup loss.

### Weak motivation why we should "decouple" lambda and the objective function

I presume that the motivation of this paper is removing (or reducing) the dependency of $\lambda$ and the objective function. If the main motivation of this paper is "addressing label mismatch", then there should be more explanations of how removing the dependency of $\lambda$ and the objective function ("decoupling") and label mismatch are related. More specifically, it is not clear to me how "decouple" fixes the label mismatch problem.

Moreover, the proposed method still uses the original loss function. What happens if we only use $L_{DM(CE)}$ alone without $L_{MCE}$? If using $L_{DM(CE)}$ alone is worse than using $L_{MCE}$ alone, then I think it is hard to say that the proposed $L_{DM(CE)}$ itself is effective.

Furthermore, to argue that the "decoupling" (ignoring the other mixing targets) is effective, then I think this paper should include the following baseline:

- `CrossEntropy(target_1, prediction)` + `CrossEntropy(target_2, prediction)` => removing $\lambda$ in the objective function

It is the same as "Complete-label CutMix" in Table 8 of CutMix paper, and it is known to be worse than the $\lambda$ mixing strategy (following Table 8 in the CutMix paper). I think this result can make the submission stronger.

### Minor

- It seems that the title "data-efficient" and the motivation are not well-matched. I suggest using a different term rather than data-efficient in the title and Section 3.
- $l$ of Equation (2) is not defined. Please fix the equation.
- Suggestions for a better formatting
    - Please specify the full terminology of AS (page 5). I presume AS stands for "Asymmetrical strategy".
    - I would like to recommend inserting "improvements" in the tables (e.g., as Table 1 of CutMix paper).
    - Please do not use `\vspace{-XX}` between sections (e.g., page 5 "3.2 Asymmetrical strategy")
    - Please do use `\vspace{XX}` between a caption and a table (I usually recommend adding a space `~` between a caption and a table)
    - Please make more spaces (e.g., `\hfill`) between Table 2 and 3 (as well as Table 4 and 5).

**Summary Of The Paper:**

This paper proposes a new objective function for mixed sample data augmentation (MSDA). Technically, the proposed Decoupled Mixup (DM) loss is equivalent to the summation of the original mixup loss and the additional losses that only consider one mixed target label while the other mixed target labels are ignored. For example, assuming a mixed label [0.7, 0.3, 0, 0] and a prediction probability [0.5, 0.45, 0.05, 0], then the DM loss is equivalent to `MixedCrossEntropy([0.7, 0.3, 0, 0], [0.5, 0.45, 0.05, 0])` (the original mixup loss) + `CrossEntropy([1, 0, 0], [0.91, 0.09, 0])` + `CrossEntropy([1, 0, 0], [0.9, 0.1, 0])` (called as "decoupled" mixup loss). The motivation of the proposed method is that the objective function of MSDA is highly correlated to the mixing ratio $\lambda$: by ignoring the other mixed targets, the DM loss is "decoupled" with the mixing ratio $\lambda$. The extensive study on image classification tasks (supervised and semi-supervised) shows that the proposed method is effective.

**Summary Of The Review:**

This paper shows great empirical improvements on various benchmarks. The extensive studies are impressive and show consistent improvements. As my comments in "Strength", this paper will be helpful for many practitioners working on ImageNet-1K.

However, I think this paper has a large room for improvement in terms of academic publication. Especially, I think this paper needs a connection between motivation and the proposed method. I cannot find any reason why the proposed method will work. In terms of the label mismatch problem, I think this method will still suffer from the label mismatch problem because the label mismatch problem can be solved only by making the mixed sample and the mixed label dependent. Furthermore, I think this paper should compare using $L_{DM(CE)}$ alone as well. If using $L_{DM(CE)}$ alone does not work well, then it means that we need a better explanation (and motivation) of the effect of the combination of $L_{MCE}$ and $L_{DM(CE)}$, not only for $L_{DM(CE)}$.

Overall, I think this paper does not reach the acceptance threshold of the ICLR conference.

----

During the reviewer-author discussion period, the authors provide additional experiments to resolve my concerns. However, as my last comment, I am still skeptical about revising my score because my concerns still remain. I will maintain my initial recommendation.

---

> ### Author Response · Authors · 2022-11-09
> **Response to Reviewer A9VU**
>
> Thank you for your precious time and great efforts. Your professional suggestions and questions enlighten us to polish the paper. We will address your questions one by one and make the corresponding changes in the reversion.
>
> > **Q1**: I cannot find any connection between the "label mismatching problem" and the "decoupled Mixup loss".
>
> The same problem of label mismatch, dynamic methods choose to spend more time to optimize the data-level mixing strategy (retaining the discriminative feature of the original sample) to alleviate the problem; while the decoupling mechanism treats the samples of label mismatch as hard samples that are not fully utilized: by interfering with the distribution of predicted labels to achieve model-autonomous discriminative feature mining.
>
> Taking CutMix in Fig.1 as an example, although most of the panda-related features are lost in the mixed sample, we should encourage the model to make as confident a prediction as possible under this condition of insufficient information, i.e., to mine the hard features. And vice versa, when small lambda corresponds to salient features (e.g., the head of a panda), the confident prediction is also suppressed by the MCE loss. Therefore, decoupled mixup is a solution from the nature of mixed samples, and dynamic methods are a solution when there is a label mismatch, but we think that the decoupled approach is simpler and more effective. We have added additional descriptions in the revised version; please ref to Section 1.
>
> > **Q2**: I cannot find any connection between "data-efficient" (the title) and the proposed method.
>
> Since the decoupled mechanism does not introduce any additional computational overhead and can steadily improve the training efficiency of mixup training and the downstream task performance in various task environments. Therefore, the decoupled mixup is data-efficient. We have strengthened the connection between DM and data-efficient in the revised paper( ref to Sections 1 and 4.4 (4)). Additionally, we provide an analysis of data-efficient mixup in Appendix A.5; please ref to the revised paper for the details.
>
> > **Q3**: Furthermore, to argue that the "decoupling" (ignoring the other mixing targets) is effective, then I think this paper should include the following baseline:CrossEntropy(target_1, prediction)+CrossEntropy(target_2, prediction)
>
> Thanks for your suggestions. In fact, we have included the results of this experiment in Table. 3 in the original paper, where MBCE(two) means using two-hot as the label for mixup training. From the table, we can clearly see that such an approach will completely lose the original smoothness nature of the mixup, which will make the overall performance drop significantly.
>
> We look forward to hearing your professional feedback and insights, and if you have any other questions, we welcome you to ask them at any time.

---

> > ### Comment · Reviewer_A9VU · 2022-11-21
> > **Response**
> >
> > Thanks for your response and sorry for the late reply.
> >
> > As my initial review, I think the major weakness of this paper is the quality of the paper. For example:
> > - "this paper needs a connection between motivation and the proposed"
> > - "I cannot find any reason why the proposed method will work"
> > - "I think this method will still suffer from the label mismatch problem because the label mismatch problem can be solved only by making the mixed sample and the mixed label dependent"
> > - "there is no baseline using DM(CE) alone"
> >
> > Although the response targeted to address my concerns, I feel that the response and the revised paper cannot address my concerns well.
> > From the response for reviewer 5tSp, I found the revised logic flow is:
> > > The logic chain can be summarized as follows:
> > > label mismatch→dynamic methods→construct discriminative features→DM helps mixup training to mine discriminative features
> >
> > I cannot agree with "DM helps mixup training to mine discriminative features". As reviewer 5tSp's comment, "the authors should provide any empirical or theoretical backups of whether this is actually addressed through the proposed DM loss.". However, after reading the whole response, I am still not convinced about the argument. Recall the argument by the authors:
> >
> > > Taking CutMix in Fig.1 as an example, although most of the panda-related features are lost in the mixed sample, we should encourage the model to make as confident a prediction as possible under this condition of insufficient information, i.e., to mine the hard features. And vice versa, when small lambda corresponds to salient features (e.g., the head of a panda), the confident prediction is also suppressed by the MCE loss. Therefore, decoupled mixup is a solution from the nature of mixed samples, and dynamic methods are a solution when there is a label mismatch, but we think that the decoupled approach is simpler and more effective
> >
> > I cannot agree with these new arguments because
> > - (1) It is unclear the connection between "we should encourage ... insufficient information" and "to mine the hard features". How the hard feature mining can solve the problem? Even if it can solve, as far as the reviewer understood, the proposed method is not related to the hard feature mining
> > - (2) The proposed method is a mixture of MCE and CE(DM); hence, it still uses MCE which suffers from the problem raised by the authors. How the additional CE(DM) term can alleviate the problem?
> > - (3) "Therefore, decoupled mixup is a solution from the nature of mixed samples, and dynamic methods are a solution when there is a label mismatch" => it is hard to agree because I cannot agree with the previous sentences (also, I still cannot find connections between the arguments).
> >
> > As my initial concerns still remain, I will maintain my initial recommendation for now.

---

> > > ### Author Response · Authors · 2022-11-23
> > > **Point-to-point Response [3/3]**
> > >
> > > > Using DM(CE) loss alone.
> > >
> > > We thank Reviewer A9VU for pointing out this ablation experiment. **However, due to the mixup only applied in the training phase, using DM(CE) alone is not a reasonable baseline or training setting.** Using DM(CE) alone will cause the inconsistency between training and testing Because DM(CE) changes the way Softmax is calculated, i.e., only considering C-1 classes to compute Softmax. This is also verified by the following ablation results on CIFAR-100 based on ResNet-18 as follows. In this table, we find that using DM(CE) alone results in degenerated performances than the original MCE.
> > >
> > > | Method | Mixup | CutMix | ManifoldMix | FMix | ResizeMix | SaliencyMix | PuzzleMix | AutoMix | SAMix |
> > > |---|:---:|:---:|:---:|:---:|:---:|:---:|:---:|:---:|:---:|
> > > | MCE | 79.12 | 78.17 | 80.35 | 79.69 | 80.01 | 79.12 | 81.13 | 82.04 | 82.30 |
> > > | DM(CE) only | -2.35 | -1.87 | -3.10 | -1.94 | -2.27 | -1.83 | -2.36 | -2.05 | -2.19 |
> > > | DM(CE) | +1.32 | +1.22 | +0.60 | +0.31 | +0.05 | +0.16 | +0.21 | +0.28 | +0.10 |

---

> > > ### Author Response · Authors · 2022-11-23
> > > **Point-to-point Response [2/3]**
> > >
> > > > Provide any empirical or theoretical backups of whether this is actually addressed through the proposed DM loss.
> > >
> > > Following [1], we conduct an occlusion evaluation with random patch drops to verify the proposed DE loss can help static and dynamic mixup methods to explore more discriminative features. Specifically, given an input image $x\in \mathbb{R}^{3\times H\times W}$ and its corresponding label $y\in \mathbb{R}^{C}$, where $C$ is the class number, the classification network $f: x\rightarrow x$ is considered robust (i.e., learning more discriminative features) if the outputs are correct label when taking an occluded image $x'$ as the input, namely $y’ = f(x')$. For occlusion, we consider patch-based random masking, and we split the image of size $224\times224$ into $N$ patches of $16\times 16$ and randomly mask $M$ patches out of the $N$ patches. The occlusion ratio is defined as $r=\frac{M}{N}$.
> > >
> > > - Random PatchDrop Evaluation based on ResNet-50 (RSB A3 setting) on ImageNet-1k: we compare RSB (MCE) with/without DMCE and PuzzleMix (MCE) with/without DMCE in the following table, whose plot is provided by an anonymous tink https://user-images.githubusercontent.com/44519745/203444951-f131d0b3-82d1-45bc-968c-a6eabe168f95.png. Note that RSB denotes using Mixup ($\alpha=0.1$) and CutMix ($\alpha=1$). We find that the proposed DMCE loss significantly improves the robustness of static and dynamic mixup methods with ResNet-50, indicating that the model captures more salient information and discriminative features that are robust to occlusion.
> > >
> > > | Method | 0\% | 10\% | 20\% | 30\% | 40\% | 50\% | 60\% | 70\% | 80\% | 90\% |
> > > |---|:---:|:---:|:---:|:---:|:---:|:---:|:---:|:---:|:---:|:---:|
> > > | RSB(MCE) | 76.49 | 73.56 | 68.13 | 58.45 | 46.31 | 32.08 | 19.57 | 10.46 | 5.29 | 1.93 |
> > > | **RSB(MCE)+DMCE** | +1.23 | +2.19 | +4.75 | +9.51 | +7.11 | +5.02 | +4.15 | +7.18 | +4.42 | +2.10 |
> > > | PuzzleMix(MCE) | 77.27 | 75.16 | 72.04 | 66.51 | 57.62 | 46.90 | 35.53 | 23.82 | 14.37 | 7.13 |
> > > | **PuzzleMix(MCE)+DMCE** | +0.33 | +1.45 | +1.60 | +2.46 | +3.43 | +3.56 | +4.65 | +4.90 | +4.08 | +2.21 |
> > >
> > > - Random PatchDrop Evaluation based on DeiT-S (DeiT setting) on ImageNet-1k: we then compare Mixup (MCE) with/without DMCE and PuzzleMix (MCE) with/without DMCE in the following table and the anonymous tink https://user-images.githubusercontent.com/44519745/203446325-adbc9f63-a215-46b9-bfcf-0bf3acf52cfb.png. Similarly, we also find that the DMCE loss improves the robustness of static and dynamic mixup methods with DeiT-S by helping the model to mine discriminative features overlook by the original MCE loss.
> > >
> > > | Method | 0\% | 10\% | 20\% | 30\% | 40\% | 50\% | 60\% | 70\% | 80\% | 90\% |
> > > |---|:---:|:---:|:---:|:---:|:---:|:---:|:---:|:---:|:---:|:---:|
> > > | Mixup(MCE) | 79.65 | 77.89 | 76.26 | 74.34 | 71.57 | 67.78 | 62.89 | 55.05 | 42.86 | 26.34 |
> > > | **Mixup(MCE)+DMCE** | +0.39 | +0.94 | +1.03 | +1.31 | +2.47 | +3.69 | +3.69 | +4.88 | +5.46 | +4.55 |
> > > | PuzzleMix(MCE) | 79.84 | 78.53 | 77.61 | 76.37 | 74.92 | 72.85 | 69.98 | 65.73 | 58.75 | 42.14 |
> > > | **PuzzleMix(MCE)+DMCE** | +0.41 | +0.98 | +0.96 | +1.15 | +1.34 | +1.64 | +2.13 | +2.89 | +3.69 | +4.73 |
> > >
> > > [1] Naseer M M, Ranasinghe K, Khan S H, et al. Intriguing properties of vision transformers. NeurIPS 2021.

---

> > > ### Author Response · Authors · 2022-11-23
> > > **Point-to-point Response [1/3]**
> > >
> > > Thanks for your great efforts and additional comments. We summarise your questions and concerns in four aspects as follows, and we will address them accordingly.
> > >
> > > > The connection between Motivation and Method and why the proposed method will work (still using MCE)?
> > >
> > > As stated in the revised paper, we have two motivations for proposing decoupled mixup:
> > >
> > > - **Motivation (1).** Ref to Sec. 1, we argue that mixup augmentation methods should not be too complex to deploy easily. It is the dynamic methods that alleviate the label mismatch problems but lack ease of use.
> > >
> > > - **Motivation (2).** Ref to Sec. 2.1, we argue that MCE loss has flaws when dealing with label-mismatched samples. For example, when a part of the discriminative feature (small $\lambda$) is retained in the mixed sample, it is not reasonable to suppress the confidence of the corresponding prediction.
> > >
> > > Therefore, we improve mixup by following two aspects: (1) We do not choose to design another complex data mixing policy but to leverage the label-mismatched samples as hard samples without any extra computation. (2) We choose to give MCE the ability to take full advantage of mixed samples by introducing a new decouple term. Specifically, MCE works fine when there are no label-mismatched samples. Once they exit, the decoupled terms enable the model to learn more discriminative features by adjusting the probability distribution of the predictions.
> > >
> > > Consider two types of mismatching when $\lambda$ is small: The corresponding feature is salient and not. The former need high confidence predictions as a matter of course; the latter need to be encouraged to mine for hard features. **Please note that these advantages from DM are built upon the smoothness property from MCE. In the next three questions, we will provide more experiments to prove that.**

---

> > > ### Author Response · Authors · 2022-12-01
> > > **Looking Forward to Your Feedback**
> > >
> > > Dear Reviewer A9VU,
> > >
> > > Hope you are going well these days. Thanks again for your insightful and detailed comments on our paper.
> > >
> > > We have updated our new responses to the raised questions and concerns of your latest comment. Meanwhile, we provided additional experiments to verify our method that can mine discriminative features overlook by the original MCE loss. Unfortunately, we cannot upload the revision now, and we will revise our paper according to your latest comment.
> > >
> > > Could you please take a look at our response, and let us know if there is any further feedback? Thank you very much!
> > >
> > > Best regard,
> > > The authors

---

> > > > ### Comment · Reviewer_A9VU · 2022-12-05
> > > > **Response**
> > > >
> > > > Thanks for your updated response. I am still skeptical about revising my score because my concerns still remain.
> > > >
> > > > > Therefore, we improve mixup by following two aspects: (1) We do not choose to design another complex data mixing policy but to leverage the label-mismatched samples as hard samples without any extra computation. (2) We choose to give MCE the ability to take full advantage of mixed samples by introducing a new decouple term. Specifically, MCE works fine when there are no label-mismatched samples. Once they exit, the decoupled terms enable the model to learn more discriminative features by adjusting the probability distribution of the predictions.
> > > >
> > > > I agree with (1). However, (2) is still not well supported. Why does the decoupled term enable the model to learn more discriminative features? It is still unclear to me (I also believe that "discriminative features" should be defined in a more rigorous way, but it could be out-of-scope in this response). It should be supported by a more reasonable explanation. For example, the CutMix paper argues that CutMix or other regional dropout methods will help to learn localizable features by preventing a model from too much focus on a very small local region on image inputs. I expect a plausible explanation like this.
> > > >
> > > > Even if the decoupled term (DM(CE)) can learn more discriminative features conceptually, as shown by the additional response, using DM(CE) alone shows worse performance than CE only. It means that DM(CE) cannot deal with the problem alone, but it still needs MCE, where the authors argue that MCE will be problematic in dealing with the problem. Therefore, I think a different logic should be required.
> > > >
> > > > Moreover, occlusion robustness cannot be evidence of the argument. As observed by [1], the current non-adversarial robustness benchmarks are highly correlated with standard accuracy. It means that if model A shows higher clean accuracy than model B, then model A will be better than model B in other robustness benchmarks as well. Showing better occlusion robustness cannot be specific evidence of (2). There should be more rigorous reasons why occlusion robustness is the evidence of "discriminative features".
> > > >
> > > > [1] Taori, Rohan, et al. "Measuring robustness to natural distribution shifts in image classification." Advances in Neural Information Processing Systems 33 (2020): 18583-18599.
> > > >
> > > > The following argument is my personal opinion, and it does not affect the decision. However, I would like to suggest the authors change their logic flow if they agree with my personal opinion.
> > > >
> > > > It could be controversial, but I don't think the label mismatch problem is a severe problem. Note that we already employ 8%-100% random crop during training. In this case, even if there is no mixup operation, "only 23.5% of the random crops have the intersection-over-union (IoU) measure greater than 50% with the ground truth boxes" [2]. In other words, 76.5% of training samples have less than 50% IoU with the ground truth boxes. If the non-mixed samples already show low IoU with the GT boxes, why the label mismatch problem is especially problematic? If the authors argue that "MCE loss has flaws when dealing with label-mismatched samples", then MCE loss also has flaws when dealing with standard training.
> > > >
> > > > [2] Yun, Sangdoo, et al. "Re-labeling imagenet: from single to multi-labels, from global to localized labels." Proceedings of the IEEE/CVF Conference on Computer Vision and Pattern Recognition. 2021.
> > > >
> > > > In my opinion, rather than arguing about the label mismatch problem, it would be better to argue about the choice of label mixing design choice: $y' = \lambda y_a + (1 - \lambda) y_b$ could not be the best option for mixing labels. Note that this is my personal recommendation for future revision.
> > > >
> > > > I encourage the authors to revise the paper to make the motivation clearer and the connection between the motivation and the proposed method clearer. I think the experimental results are extensive and the proposed method can bring a benefit for mixup methods, but this paper has a large room for improvement in terms of academic publication.
> > > >
> > > > For these reasons, I will maintain my initial recommendation.

---

### Official Review · Reviewer_5tSp · 2022-10-24

**Confidence:** 3
**Correctness:** 3
**Technical Novelty And Significance:** 2
**Empirical Novelty And Significance:** 2
**Recommendation:** 5

**Clarity, Quality, Novelty And Reproducibility:**

This paper is somewhat clearly presented, but the entire notations should be refined for a better presentation. This method seems to provide some novelty over the conventional Mixup-based methods. The reproducibility would not be a matter with the simple modification of the loss like this method.


**Strength And Weaknesses:**

Pros)
+ The idea looks simple yet practical for the practitioners using Mixup in training.
+ This paper provides extensive empirical studies to support the effectiveness of the proposed method.
+ All the experimental results consistently improve when adopting the proposed loss upon the baseline Mixup methods.

Cons)
- There is no clear evidence that decoupling the class can reduce the label mismatch. It hardly agrees that only performance improvements cannot support the claim. Instead, the authors should provide any empirical or theoretical backups of whether this is actually addressed through the proposed DM loss.
- Some notations are not clearly presented:
  - $p(a,b)$ seems to be the probability of the mixed samples; $p_a$ and $p_b$ must be scalars. I could understand it only after reading the manuscript carefully.
  - l, i, and j in eq.(2)
  - M in Section 2.2
  - $y^i_a$, $y^j_b$ in proposition 2

Comments)
- The authors claim that the proposed method can adaptively focus on discriminative features as the dynamic Mixup methods, but they seem to be conceptually different. If not, there should exist any backups for this. Presumably, the performance improvements upon the dynamic Mixup methods in Table 1 seem to support my concern.
- How does "the discriminatory of instances" in p.2 be achieved through the proposed method?
- Please explain in more detail why the regressing lambda is the problem.
- Vanishing $\lambda$ in eq.(2) release the gradient magnitude not to become having a dependency on $\lambda$, but still unclear that 1) it affects the prediction; 2) it is actually connected with the label mismatching. Please clarify this.


**Summary Of The Paper:**

This paper proposes a variant of the Mixup method that considers the label mismatch problem by decoupling (i.e., removing) the competitor's class in the softmax which can potentially cause mismatching between the mixed label and sample. The authors formulate the decoupled mixup cross-entropy loss ($L_{DM(CE)}$) that becomes an element of the proposed decoupled mixup (DM) loss  ($L_{DM}$) with the standard mixed cross-entropy loss ($L_{MCE}$). The authors claim that using the DM loss can address the label mismatch problem and argue a similar effect as dynamic mixup methods. The authors apply the DM loss on Image classification tasks on the datasets of CIFAR-100, Tiny-ImageNet, and ImageNet-1k with diverse network architectures and baseline Mixup methods. The authors apply the proposed loss not only to the static Mixup methods but to the so-called dynamic Mixup methods such as SaliencyMix, PuzzleMix, AutoMix, and SAMix. Moving further from the conventional image classification experiments, the authors further evaluate fine-tuning classification on FGVC datasets and semi-supervised learning methods. All the experimental results consistently back the effectiveness of the proposed loss.

**Summary Of The Review:**

This work provides a novel Mixup-based loss for consistent performance improvements on several datasets with many models. I like the constant improvements over diverse setups and scenarios. However, what I'm most concerned about is that there is actually novelty in the sense of a new loss of performance gains, but there doesn't seem to be enough evidence to support the author's claim. I recommend the authors revise the paper by providing more materials that show the link between the decoupling by the proposed DM loss and the label mismatch problem; consider more on the gap between embellishing the proposed method as a dynamic aspect and the ones categorized into dynamic methods.

---

> ### Author Response · Authors · 2022-11-09
> **Response to Reviewer 5tSp**
>
> Thank you for your precious time and great efforts. Your insightful suggestions and professional questions are the keys to improving the quality of the paper. We will address your questions one by one and make the corresponding changes in the reversion.
>
> > **Q1**: There is no clear evidence that decoupling the class can reduce the label mismatch. The authors should provide any empirical or theoretical backups of whether this is actually addressed through the proposed DM loss.
>
> We have made changes to the revised version to make the presentation clearer: we are not solving the label mismatch problem, but we are trying to make full use of the label-mismatched samples.
> The next answer may be a bit lengthy, but hopefully, it could bring answers to your questions.
> Different from dynamic mixup methods, we do not spend extra optimization time for the strategy of mixing data to alleviate the label mismatch problem: create "simple" mixed samples at the data level (retaining the most discriminative features).
>
> However, we find that since retaining discriminative features can improve the effectiveness of mixup training, we can also consider starting from the label level to encourage the model to mine discriminative features automatically instead of handcrafting such simple samples as dynamic methods.
>
> Specifically, taking CutMix in Figure 1 as an example, such a mixed sample for Panda can be treated as a hard mixed sample. That is, although Panda’s feature is not salient in this mixed sample, we need to encourage the model to discover the hard features associated with Panda and give more confident predictions, which is a property that MCE does not have.
> Because even if the model is able to give a prediction greater than 0.4 according to the evidence of Panda’s foot, the MCE will go ahead and suppress such a prediction due to $\lambda$.
>
> Therefore, we do not directly try to solve the label mismatch problem but introduce the decouple mechanism based on the development idea of mixup methods, hoping that it can further encourage mixup methods to mine hard features in mixed samples and give the model the ability to produce confident predictions based on MCE.
>
> The logic chain can be summarized as follows:
>
> label mismatch→dynamic methods→construct discriminative features→DM helps mixup training to mine discriminative features
>
> > **Q2**: Some notations are not clearly presented:
>
> Thanks for your precious time and careful reading. We have corrected the notations in reversed version. The meaning of $M$, $y_a$, $y_b$, and superscripts are explained at the beginning of Section 2.
>
> > **Q3** and **Q4**: The authors claim that the proposed method can adaptively focus on discriminative features as the dynamic Mixup methods, but they seem to be conceptually different. How does "the discriminatory of instances" in p.2 be achieved through the proposed method?
>
> Although dynamic mixup is more straightforward in terms of implementation approach, the decoupling mechanism is a further leveraging of the mixed sample based on MCE: as stated in Q1, even when the mixed features are not salient, the DM encourages the model to identify the discriminative features and make more confident predictions in this situation. Thus, the model's ability to extract discriminative features is enhanced by leveraging label-mismatched samples.
>
> > **Q5**:  Please explain in more detail why the regressing lambda is the problem.
>
> The regression of lambda values itself is not a problem but has some drawbacks. As stated in Q1, the existence of the label mismatch problem prevents the model from taking full advantage of the mixed data. Although dynamic mixup alleviates this problem at the cost of additional computation to construct optimized samples, we believe that a similar effect can be achieved by controlling the probability distribution of the prediction. That is, we treat samples with label mismatch as hard samples and enhance the model's mining and extraction of discriminative features by computing the predicted probability of each class independently.
>
> > **Q6**: Vanishing λ in eq.(2) releases the gradient magnitude not to become having a dependency on λ, but still unclear that 1) it affects the prediction; 2) it is actually connected with the label mismatching. Please clarify this.
>
> In the static mixup, we can find a lot of mixup samples with another kind of label mismatch (different from Fig. 1): the corresponding lambda has a small value but contains salient features (like the Panda’s head). In this case, once the prediction is wrong, we would like to give harsh feedback to the model, but the magnitude is drastically reduced due to the constraint of the lambda. The introduction of DM can alleviate this situation significantly, which is why DM can bring more gain to static mixup than dynamic ones.
>
> We look forward to hearing your professional feedback and insights, and if you have any other questions, we welcome you to ask them at any time.

---

### Official Review · Reviewer_baqd · 2022-10-25

**Confidence:** 3
**Correctness:** 2
**Technical Novelty And Significance:** 3
**Empirical Novelty And Significance:** 2
**Recommendation:** 5

**Clarity, Quality, Novelty And Reproducibility:**

The paper proposed an original loss function to optimize the mixup style objective. Extensive experimental results on the supervised and semi-supervised tasks valid the effectiveness of the proposed loss function.


**Strength And Weaknesses:**

The paper proposed a new loss function, and extends to the semi-supervised case and binary cross entropy form. the experimental results show that it’s better than previous benchmarks.
Some Questions:
1. Some annotations are misleading.  like in formula (2),  where is l defined. For the asymmetrical strategy DM loss, the log(\phi) is a matrix or a vector?
2. What’s the meaning of “data-efficient”? Can you show the model performance under different sizes of dataset to prove the data-efficiency advantage over the baselines?
3. As the paper argued, one of the advantages is that it’s efficient computing, can you show experiments on how compute-efficient compared with the baselines?


**Summary Of The Paper:**

In  this paper,  to solve the expensive computing resources problem of current dynamic mixup method, it proposed a method trying to transfer the decoupling mechanism of dynamic methods from the data level to the objective function level and propose the general decoupled mixup loss. The experimental results on supervised and semi-supervised tasks proved the effectiveness of DM.


**Summary Of The Review:**

In summary, the paper proposed a novel loss function and verified the effectiveness on different dataset and different tasks.  There are some minor clarity issues on the formulations. However, the paper doesn’t provide proof on how “data-efficient” of the proposed method is and there are no Quantitative results to show the computation efficiency advantage.

---

> ### Author Response · Authors · 2022-11-09
> **Response to Reviewer baqd**
>
> Thank you for your great efforts, these valuable questions and constructive suggestions, which are exactly what the paper needs. We will take your suggestions and solve your problems one by one, and all corresponding changes will be reflected in the revision.
>
> >**Q1**: Some annotations are misleading. like in formula (2), where l is defined. For the asymmetrical strategy DM loss, the $log(\phi)$ is a matrix or a vector?
>
> In the paper, the result of $\sigma(\cdot)$ is a vector of categorical probabilities, while the result of $\phi(\cdot)$ is a matrix of categorical probabilities. We have made explanations and corrections to these issues in the revised version; please ref to Section 3.1.
>
> >**Q2**: What’s the meaning of “data-efficient”? Can you show the model performance under different sizes of datasets to prove the data-efficiency advantage over the baselines?
>
> In this paper, data-efficient means that replacing only the decoupled loss in the same mixup training environment can steadily improve the performance of the model in downstream tasks without any additional computational overhead, as we describe in detail in the revision of Section 1 and 4.4 (4).
> In fact, the efficiency is reflected in the results of our extensive experiments, especially in semi-supervised settings with limited data, such as Tables 6 and 7, where it can be seen that the less labelled data available, the greater the gain from DM.
> Moreover, according to your suggestion, we further conducted additional experiments on CIFAR-100 with different sizes of training data, as below. The results have also been added to Appendix A.5. Thanks again for your suggestion to improve the quality of the paper.
>
> |             |  15\% |    15\%   |  30\% |    30\%   |  50\% |    50\%   |
> |-------------|:-----:|:---------:|:-----:|:---------:|:-----:|:---------:|
> | Methods     |  MCE  |   DM(CE)  |  MCE  |   DM(CE)  |  MCE  |   DM(CE)  |
> | Vanilla     | 42.48 |     -     | 56.41 |     -     | 64.32 |     -     |
> | Mixup       | 42.23 | **44.39** | 55.61 | **56.78** | 64.55 | **65.92** |
> | CutMix      | 43.81 | **44.85** | 55.99 | **57.14** | 64.38 | **65.87** |
> | SaliencyMix | 42.95 | **44.01** | 55.42 | **56.51** | 64.56 | **66.10** |
> | PuzzleMix   | 42.67 | **43.87** | 56.19 | **57.36** | 64.74 | **66.26** |
>
> > **Q3**: As the paper argued, one of the advantages is that it’s efficient computing, can you show experiments on how compute-efficient compared with the baselines?
>
> Since DM loss does not bring any additional computational overhead, we can see from extensive experiments that the improvement brought by DM is consistent. Therefore, we can say that the mixup training with DM is computationally efficient.
>
> We look forward to hearing from you, and if you have any other questions, we welcome you to ask them at any time.

---

### Official Review · Reviewer_4Ujo · 2022-10-26

**Confidence:** 3
**Clarity, Quality, Novelty And Reproducibility:** See the comments above.
**Correctness:** 3
**Technical Novelty And Significance:** 3
**Empirical Novelty And Significance:** 3
**Recommendation:** 5

**Strength And Weaknesses:**

### Strength
1. The proposed method is well motivated, simple, technically sound,  of generalizability.
2. Extensively experiments are conducted to verify the proposed method across different tasks, datasets, and other alternative choices.
3. The paper is well presented, with professional figures, table and language use.

### Weaknesses
1. Setting the parameter looks very important to the performance. Sensitivity analysis has been conducted in the one dataset, which shows the parameter is important to the performance. It is unclear how the parameters are set in other tasks. Considering that in many cases the. proposed method is slightly better than the naive mixup method, the value of this method would significantly reduce if it is non-trivial to tune the parameter to get results.

2. The weakness of this work has not been discussed. As a plug-and-play tool like this, it would be highly valuable if the authors can reveal the weakness so that people can decide whether to adopt it for their own projects.

**Summary Of The Paper:**

This paper revisits the commonly used mixup technique and proposes to modify the loss function to achieve similar effect as the time-consuming search based methods. The proposed method is simple and proved effective across various tasks, including classification, semi-supervised learning on both small and large scale datasets.

**Summary Of The Review:**

Overall this paper is good: it extends an existing widely-used technique with simple modification that produces good results across various tasks. There are some issues in the experiments, which I do not think undermine the value of this work.


Post rebuttal comments: After reading the comments from other reviewers, I tend to reject this submission. While this submission indeed proposes a novel technique that generalizes well across various tasks, the technical contribution is slightly below the bar this prestigious conference. So, I would like to join other reviewers and vote for a rejection.

---

> ### Author Response · Authors · 2022-11-09
> **Response to Reviewer 4Ujo**
>
> Thank you for your effort and very constructive comments and suggestions to help us improve the quality of the paper. We will address your questions one by one and make the corresponding changes in the reversion.
>
> > **Q1**: Need comprehensive sensitivity analysis on hyper-parameters.
>
> We provide more sensitivity analyses on different datasets in Appendix A.4. For more information, please refer to the revised paper.
> What can be found is that the additional experimental observations are consistent with what we described in 4.4(1): static and dynamic mixup policies have their preferences for hyperparameters, and their values are very stable, e.g., eta prefers smaller values (0.1) in the static method and larger values (1.0) in the dynamic one.
>
> > **Q2**: Weakness discussion.
>
> Decoupled mixup (DM) as a plug-and-play method can steadily improve the performance of downstream tasks without the additional computational overhead.
> However, one of the obvious drawbacks of DM is the introduction of additional hyperparameters.
> In conjunction with your suggestion (Q1), the additional experiments we have done indicate that the hyperparameters are stable. However, if the user applies it to data other than images, it may take some extra time to check whether the default hyperparameters are reliable. In general, we default to the CE version of decoupled mixup, which has only one adjustable parameter, so the trial and error cost is low. We added the weakness discussion in Section 6; please ref to the revised paper.
>
> We are looking forward to your response, and if you have any further questions, please feel free to ask.

---

### Author Response · Authors · 2022-11-09
**General Response**

We sincerely thank all reviewers for their thoughtful and constructive reviews of our manuscript. We are encouraged to hear that **all** reviewers find our **method** well-motivated, simple yet practical, and easy to adapt to any mixup methods. **All** of them also think our **experiments** are extensive and comprehensive to demonstrate the consistent improvements of the proposed method across various tasks and datasets. In response to feedback, we provide responses to address the concerns of each reviewer point by point and upload a revised manuscript. The changes made in the revision are highlighted in $\color{red}{red}$. Please let us know if you have any further questions. Thanks again for all the efforts and help!

---

### Author Response · Authors · 2022-11-16
**Look forward to post-rebuttal feedback**

Dear Reviewers,

Thanks again for your insightful and helpful comments! We have responded to questions and concerns point by point and uploaded the revised manuscript. Since the discussion deadline is approaching, please let us know if you have further questions. Look forward to your further reply!

Best regards,
Authors.

---

### Decision · Program_Chairs · 2023-01-20

**Decision:**

Reject

**Justification For Why Not Higher Score:**

- Method is not well-motivated and justified in spite of its strong empirical performance


**Justification For Why Not Lower Score:**

N/A

**Metareview: Summary, Strengths And Weaknesses:**

This paper proposes a new mixup-based loss for data augmentation. Reviewers appreciated the generality of the proposed method in that it can be used with existing mixup-style methods, and the extensive experiments showing its effectiveness. The major concerns were about the weak logical link between the stated motivation of resolving "label mismatch" and the proposed method. There was an extensive discussion on this point with the authors, but reviewers remained unconvinced. Reviewers lean towards rejection overall. The AC agrees with the concerns regarding the motivation and justification of the method in spite of its strong empirical performance. The authors are encouraged to consider the reviewers' comments and strengthen the paper accordingly for a future submission.